# Mating activates neuroendocrine pathways signaling hunger in *Drosophila* females

Meghan Laturney, Gabriella R Sterne, Kristin Scott*

University of California, Berkeley, Berkeley, United States

**Abstract** Mated females reallocate resources to offspring production, causing changes to nutritional requirements and challenges to energy homeostasis. Although observed across species, the neural and endocrine mechanisms that regulate the nutritional needs of mated females are not well understood. Here, we find that mated *Drosophila melanogaster* females increase sugar intake, which is regulated by the activity of sexually dimorphic insulin receptor (Lgr3) neurons. In virgins, Lgr3+ cells have reduced activity as they receive inhibitory input from active, female-specific pCd-2 cells, restricting sugar intake. During copulation, males deposit sex peptide into the female reproductive tract, which silences a three-tier mating status circuit and initiates the female postmating response. We show that pCd-2 neurons also become silenced after mating due to the direct synaptic input from the mating status circuit. Thus, in mated females pCd-2 inhibition is attenuated, activating downstream Lgr3+ neurons and promoting sugar intake. Together, this circuit transforms the mated signal into a long-term hunger signal. Our results demonstrate that the mating circuit alters nutrient sensing centers to increase feeding in mated females, providing a mechanism to increase intake in anticipation of the energetic costs associated with reproduction.

## Editor's evaluation

After mating, animals show a repertoire of behavioural changes. In flies, this includes an increase in egg-laying, salt, and food (particularly protein) consumption, and a concomitant decrease in sexual receptivity. This valuable study compellingly shows that flies also have an increased sugar appetite and they identify the central brain circuitry that controls this increase in the mated condition.

*For correspondence:
kscott@berkeley.edu

Competing interest: The authors declare that no competing interests exist.

## Introduction

Animals choose what to eat. Their choices are reflections of their physiological demands as they seek out and consume food to meet their metabolic needs. Beyond food deprivation, other states can impose new nutritional requirements, shifting nutrient intake to maintain homeostasis. Female mating status, for example, impacts feeding decisions across species. In *Drosophila*, mated females increase egg production while also escalating nutrient consumption. Observed diet-related changes include an increase in total food intake (*Carvalho et al., 2006*) and the development of appetites for protein and salt (*Vargas et al., 2010*; *Ribeiro and Dickson, 2010*; *Walker et al., 2015*). These two nutrients have been linked to reproduction as dietary protein and salt are likely processed and used during egg assembly (*Simpson et al., 2015*; *Walker et al., 2015*). Thus, mated females seek out food rich in specific nutrients to couple feeding behavior with metabolic demand.

Another nutrient, sugar, may also be a vital postcopulatory diet component. Sugar is the fly's main energy source (*Simpson et al., 2015*). Females use sugar during egg production when dietary carbohydrates are synthesized into lipids (*Brown et al., 2022*) and packed into the developing ova (*Sieber*

*and Spradling, 2015*). Mated females also increase their locomotor activity (*Isaac et al., 2010*), resulting in elevated metabolic rates (*Brown et al., 2022*) and driving the need for additional calories. Together, this predicts that mated females require significantly more sugar than virgins. However, as excessive sugar negatively impacts fly health (*Baenas and Wagner, 2022*), sugar intake must be tightly regulated. Although the sugar-to-protein ratio influences reproductive output and longevity (*Simpson et al., 2015*), the absolute levels of sugar intake of mated females remain untested. Moreover, how the mated state impinges upon neural and/or endocrine systems to modify feeding in mated females is not yet known.

In *Drosophila*, mated-related changes in female behavior and physiology are orchestrated by sex peptide (SP). SP is a 36-amino acid peptide that is produced in the male seminal fluid and transferred to females during copulation (*Chen et al., 1988*; *Liu and Kubli, 2003*). SP binds to its receptor (SPR; *Yapici et al., 2008*) which is expressed in sensory neurons (sex peptide sensory neurons, SPSNs) in the uterus (*Häsemeyer et al., 2009*; *Yang et al., 2009*). SPSNs convey mating status to sex peptide abdominal ganglia (SAG) neurons that ascend to the central brain (*Feng et al., 2014*) and synapse onto pC1 neurons (*Wang et al., 2020*). This female-specific SPSN-SAG-pC1 circuit is active in virgin females and silenced after mating (*Feng et al., 2014*; *Wang et al., 2020*). Reception of SP and the consequential silencing of this circuit initiates the long-term postmating response, an umbrella term used to describe the shift in many behaviors after mating including reduced sexual receptivity and increased egg laying (*Feng et al., 2014*; *Wang et al., 2020*). Although the three-tier SPSN-SAG-pC1 circuit appears to coordinate postmating state (*Walker et al., 2015*; *Feng et al., 2014*; *Wang et al., 2020*), different neurons downstream of pC1 independently adjust individual behaviors (*Wang et al., 2020*; *Wang et al., 2021*).

Similar to other postmated behaviors, mated-related changes in feeding are also regulated by SP and the activity of this circuit. The reception of SP during mating or the artificial silencing of the first-order SPSNs and second-order SAG neurons causes an increase in the consumption of both salt and protein (*Ribeiro and Dickson, 2010*; *Walker et al., 2015*). This demonstrates that the mating status circuit alters food consumption in mated females to couple nutritional intake with internal needs. But how mating status is integrated into circuits that modulate feeding is unclear. For example, with the use of whole brain imaging, researchers have identified the 'borboleta' region as a modulator of yeast-based feeding in mated females. However, if and how this region is regulated by the SPSNs, SAG, or pC1 remains unknown (*Münch et al., 2022*). Moreover, it is not yet determined if the mating status circuit also regulates the intake of other nutrients such as sugar that may be of vital use to mated females. Thus, how mating influences nutritional state circuits or feeding circuits remains a central question.

Here, we investigated how mating status influences sucrose intake. We used automated behavioral assays, powerful genetic tools, connectomics, and functional imaging approaches to examine neural mechanisms for appetite changes in mated females.

## Results

### Females increase sugar intake after mating via changes in feeding microstructure

To investigate the impact of female mating status on sugar intake, we monitored individual feeding bouts over time using a high-throughput, automated feeding platform (FLIC; *Ro et al., 2014*; *Figure 1A*). We compared the consumption of virgins and two types of mated females (1 hr or 72 hr postmated). We found that 72 hr postmated females consumed for a significantly longer duration on sucrose solution than virgins or 1 hr postmated females (*Figure 1B*). To further evaluate the onset of increased sucrose consumption, we monitored the consumption of virgin, 24 hr postmated females, and 72 hr postmated females and established a postmated phenotype in both mated groups (*Figure 1C*). These studies demonstrate that mated females consume more sugar than virgins and that changes in sucrose consumption manifest 6–24 hr after copulation.

To explore differences in feeding dynamics, virgin and mated females were allowed to feed on different sucrose concentrations and the number and length of feeding bouts were examined. Mated females consumed for longer durations than virgins for all sucrose concentrations below 500 mM (*Figure 1D*). Investigation into the microstructure of feeding revealed that the postmated increase

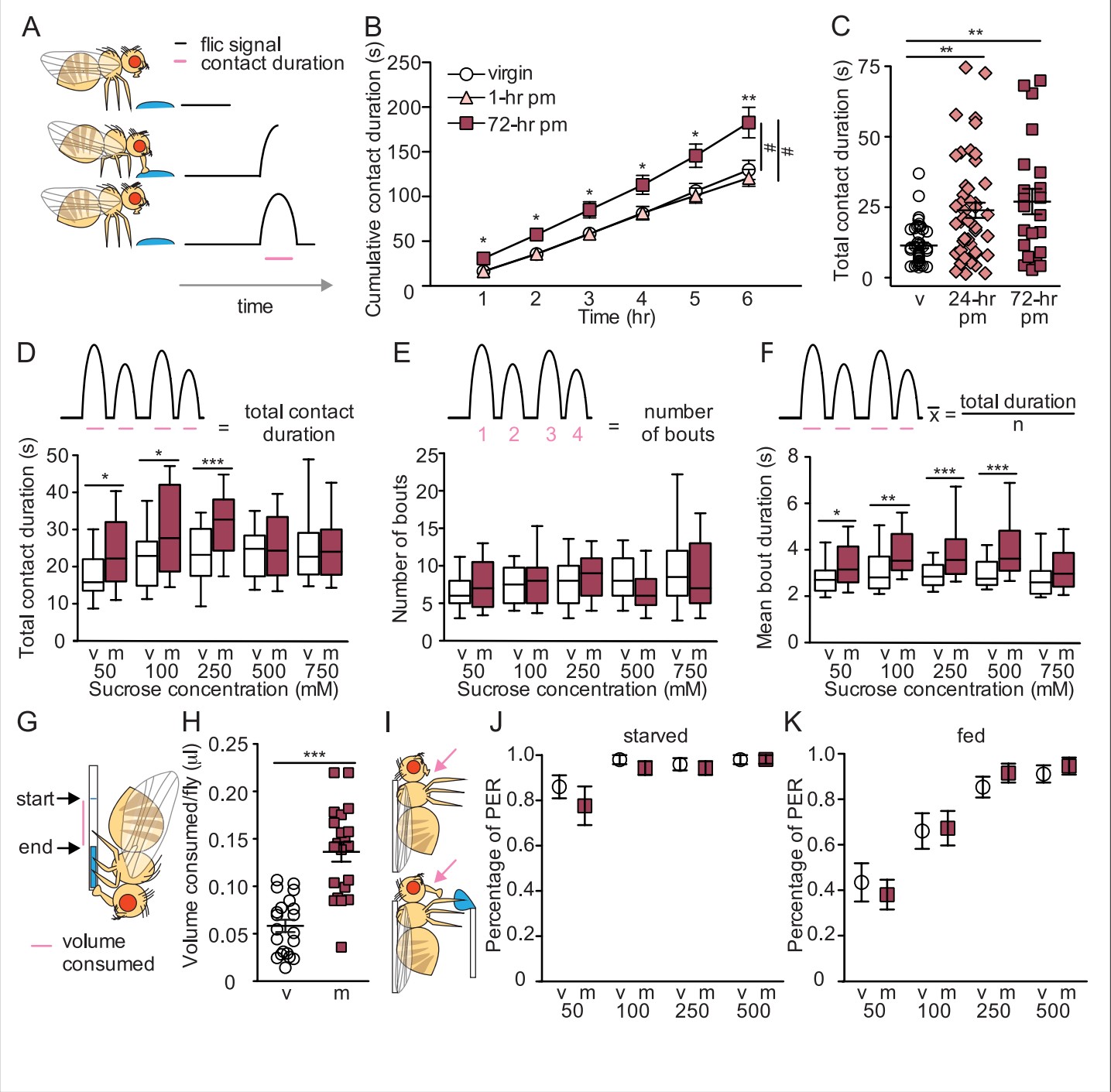

**Figure 1.** Mated females consume more sucrose than virgins by elongating feeding events. (**A**) Schematic of fly feeding behavior (left) and the fly liquid interaction counter (FLIC) signal that monitors food contact over time (right). (**B**) Cumulative drinking time of 50 mM sucrose in FLIC of the virgin (white circles, n=34), 1 hr postmated (pm) (light pink triangles, n=30), and 72 hr pm (dark pink squares, n=28) females. Line graph shows mean and s.e.m. over time. Kruskal-Wallis was used to compare groups at hour intervals, Dunn's post hoc at 6 hr, #p<0.05. (**C**) Total drinking time of 50 mM sucrose in FLIC in the 20 min assay of virgin (v; n=47), 24 hr pm (n=47), and 72 hr pm (n=22) females, Kruskal-Wallis, Dunn's post hoc. (**D–F**) Schematic and plots of feeding behavior of virgin (v) and mated (m) females presented with sucrose of varying concentrations in FLIC 20 min assay (n=33–36) examining total drinking time (**D**), number of drinking bouts (**E**), and average bout duration (**F**). Box plots show first to third quartile, whiskers span 10–90 percentile, Mann-Whitney. (**G**) Schematic of female drinking from capillary containing sucrose solution in CAFE assay. (**H**) Total volume consumed of 50 mM sucrose in CAFE assay per fly in 24 hr by virgin (v, n=21) and mated females (m, n=21), unpaired t-test. (**I**) Schematic of proboscis extension response (PER) assay. (**J and K**) Percentage of proboscis extensions observed per fly upon three presentations of sucrose of indicated concentration (mM) in virgin (v) and mated

*Figure 1 continued on next page*

*Figure 1 continued*

(m) females in starved, n=17–18 (**J**) and fed state, n=16 (**K**), Mann-Whitney. *p<0.05, **p<0.01, ***p<0.001. Scatter plot shows mean +/− s.e.m. Column graph show mean +/− s.e.m. See *Supplementary file 1* for full genotypes.

in sucrose consumption is due to an elongation in feeding bout duration rather than an increased number of bouts (*Figure 1E and F*). Thus, mated females consume more by engaging in longer feeding times rather than by initiating more feeding events.

We tested two predictions suggested by the mating-induced changes in sugar-feeding dynamics. First, longer overall feeding durations suggest that a greater amount of sugar is consumed. We directly tested this using a capillary feeder (CAFE; *Ja et al., 2007*; *Figure 1G*) and found that mated females consume significantly more sucrose than virgins (*Figure 1H*). Second, virgin and mated females execute a similar number of feeding bouts, suggesting that females do not differ in the probability of initiating a feeding event. To directly test this, we monitored proboscis extension upon sugar taste detection (*Figure 1I*) and found that virgin and mated females have similar feeding initiation propensities (*Figure 1J and K*). These results further corroborate that increased sugar consumption in mated females is due to longer meal duration rather than more frequent feeding bouts.

Hunger increases the sensitivity of gustatory neurons to sugar (*Inagaki et al., 2012*). To test whether changes in sugar sensory sensitivity underlie increased sucrose consumption in mated females, we monitored taste-induced neural activity in sugar gustatory neurons in mated and virgin females with the GCaMP6s calcium indicator (*Figure 2A*). Stimulation of the fly proboscis with sugar elicited similar responses in gustatory axons regardless of mating status (*Figure 2B–D*), demonstrating that gustatory sensitivity is not altered by mating and not likely to contribute to changes in feeding.

## Postmated increases in sucrose consumption are initiated by the mating status circuit

As mating increases egg production, we reasoned that the increase in sucrose consumption could be driven by a need-dependent mechanism. In this scenario, mating induces egg production, depleting energy stores and driving a need for sucrose consumption to restore homeostasis. To test this, we quantified sucrose consumption of eggless virgin females and eggless mated females with both FLIC (hs-bam, *Figure 3A*) and CAFE (ovoD, *Figure 3—figure supplement 1A*). We found that despite a lack of egg production (*Walker et al., 2015*; *Ohlstein and McKearin, 1997*; *Mével-Ninio et al., 1996*), mated females consumed more sucrose than virgins, demonstrating that the change in postmated

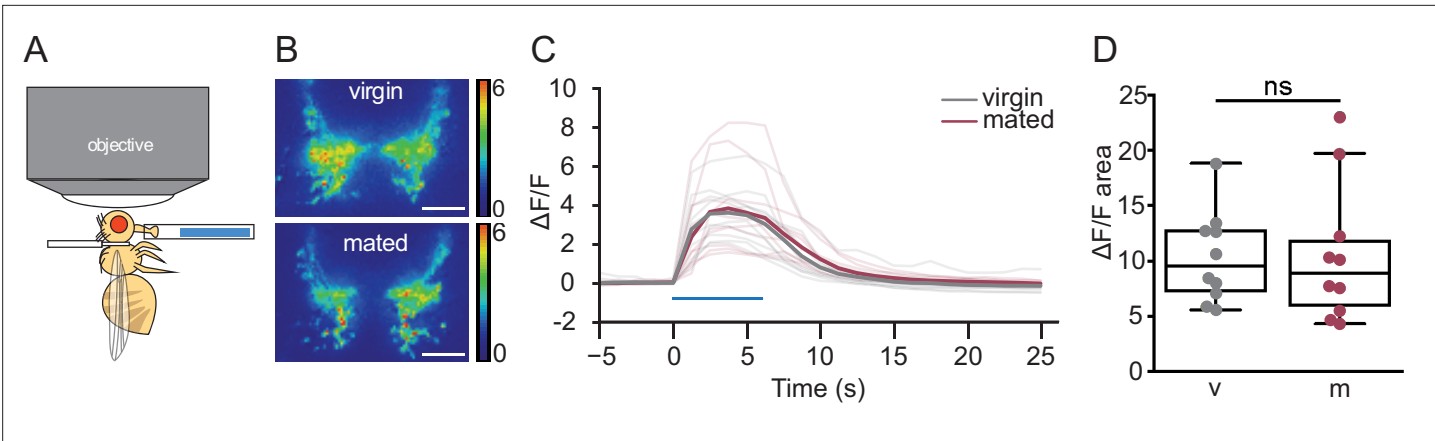

**Figure 2.** Mated and virgin females do not differ in taste-induced sugar gustatory sensory neuronal activity. (**A**) Schematic of in vivo calcium imaging experiment monitoring taste-induced activity in gustatory axons. (**B–D**) Changes in GCaMP6s signal in virgin (gray) and mated (dark pink) females upon presentation of 250 mM sucrose, shown as representative fluorescent changes in single brains (**B**), changes over time (**C**), and area under the curve (**D**), n=10 per group. Blue bar represents proboscis sucrose stimulation, Wilcoxen Rank test. ns = not significant. Scatter plot shows raw data. Box and whisker plot shows median and quartiles. Scale bar 25 µm. See *Supplementary file 1* for full genotypes.

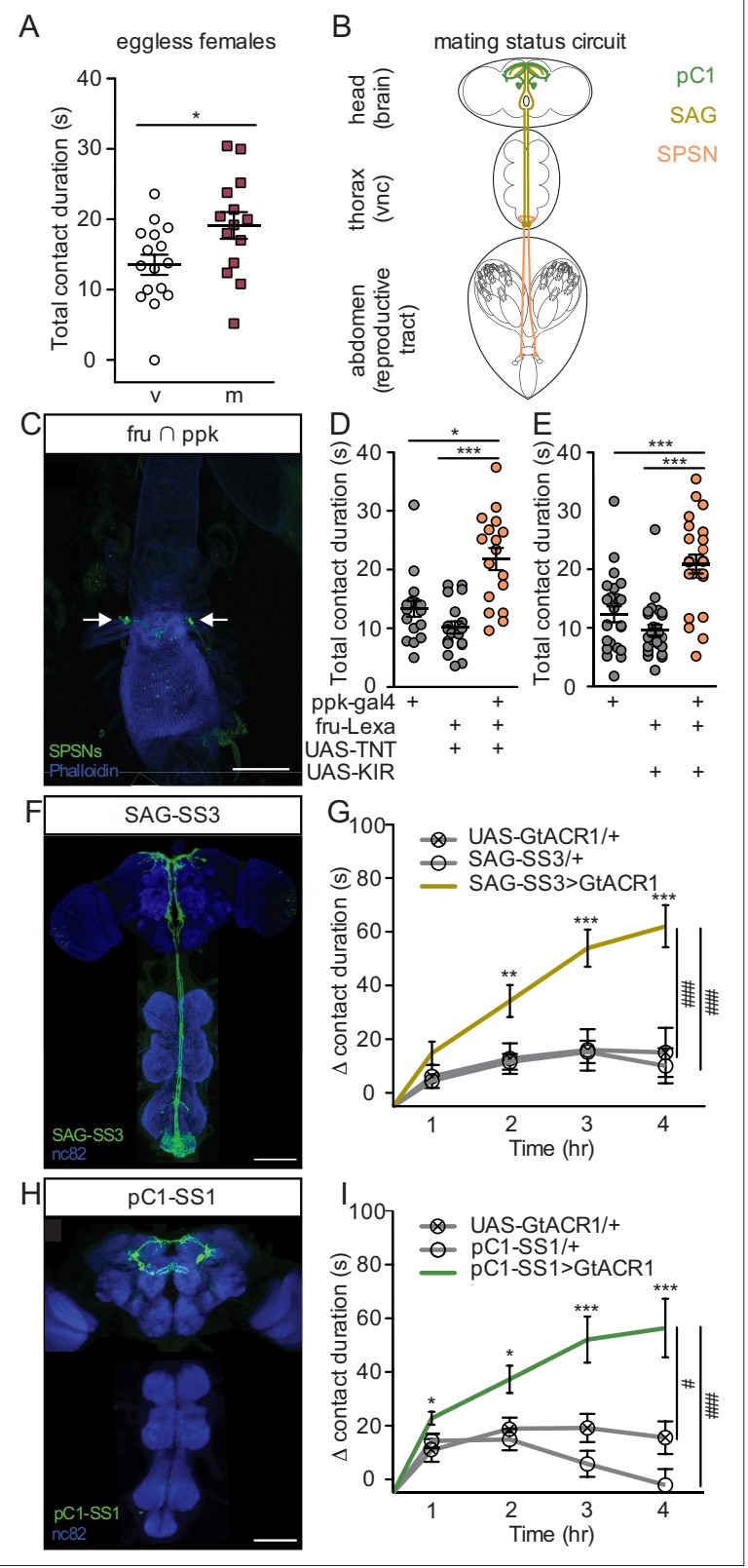

**Figure 3.** Postmated increases in sucrose consumption are induced by the canonical mating status circuit and are independent of egg-production. (**A**) Total drinking time of 250 mM sucrose in fly liquid interaction counter (FLIC) in the 20-min assay of the eggless virgin (v, n=16) and eggless mated (m, n=15) females, Mann-Whitney. (**B**) Schematic of the female with the mating circuit, showing first order (sex peptide sensory neurons, SPSNs,

*Figure 3 continued on next page*

*Figure 3 continued*

pink-orange), second order (sex peptide abdominal ganglia, SAG, dark yellow), and third order (pC1, green) neurons. (**C**) Confocal image of uterus stained to reveal SPSNs (UAS-CD8:GFP, green, arrow indicating cell bodies) and muscle tissue (phalloidin, blue). (**D and E**) Total drinking time of 250 mM sucrose in FLIC in the 20-min assay of virgin females of indicated genotype (n=17–28), Kruskal-Wallis, Dunn's post hoc. (**F and H**) Confocal images of the brain (top) and ventral nerve cord (bottom) of SAG-SS3 (**F**) and pC1-SS1 (**H**) females, stained to reveal neural projection pattern (UAS-mVenus, green) and all synapses (nc82, blue). (**G and I**) Drinking time of 250 mM sucrose by females exposed to light normalized to dark controls of indicated genotype in FLIC over 4 hr (each data point is the drinking time of an individual in the light condition minus the daily average drinking time of all females in dark condition within a genotype, n=17–39), Kruskal-Wallis, Dunn's post hoc at 4 hr, #p<0.05, ###p<0.001. Line graphs show mean and s.e.m. over time. *p<0.05, **p<0.01, ***p<0.001. Scatter plot shows mean +/− s.e.m. Scale bar 50 μm. See *Supplementary file 1* for full genotypes. See also *Figure 3—figure supplement 1*, *Figure 3—figure supplement 2*, and *Figure 3—figure supplement 3*.

The online version of this article includes the following figure supplement(s) for figure 3:

**Figure supplement 1.** Three independent SPSN lines regulate postmated sucrose consumption.

**Figure supplement 2.** Contact duration times of virgin females upon optogenetic manipulation of neurons in the mating status circuit: sex peptide abdominal ganglia (SAG) and pC1.

**Figure supplement 3.** An additional neuron in the SAG-SS3 line, SAGb, does not influence sucrose consumption.

sucrose feeding is not driven by egg production itself. Instead, this result suggests that sucrose feeding changes are anticipatory in nature and a consequence of a need-independent mechanism.

Postmated changes in female behavior are elicited by SP (*Chen et al., 1988*; *Liu and Kubli, 2003*; *Laturney and Billeter, 2015*), a male-derived seminal fluid peptide, and require female uterine primary sensory neurons (SPSNs; *Yapici et al., 2008*; *Häsemeyer et al., 2009*; *Yang et al., 2009*; *Figure 3B*). To test the involvement of the SPSNs in postmated sucrose feeding changes, we genetically accessed SPSNs using three different genetic driver combinations (see methods). All genetic intersections consistently labeled at least four SPSNs (*Figure 3C*, *Figure 3—figure supplement 1B–1C*) with no overlap of off-target neurons (*Figure 3—figure supplement 1D*), making them useful tools to investigate the role of the SPSNs in postmating feeding changes. We silenced the SPSNs in virgin females using constitutive silencers (either the potassium inward rectifier, KIR2.1, or the tetanus toxin light chain, TNT) to mimic the mated state and measured sucrose consumption. Virgin females with SPSNs silenced consumed significantly more sucrose than virgin controls (*Figure 3D and E*, and *Figure 3—figure supplement 1E–1F*), recapitulating the mated phenotype. This data argues that mated females increase sucrose consumption after mating via the SPSNs.

The canonical sex peptide pathway for postmated changes in female behavior is a three-layer circuit from SPSNs to SAGs to pC1 (*Feng et al., 2014*; *Wang et al., 2020*; *Wang et al., 2021*). As with the SPSNs, SAG, and pC1 neurons are active in a virgin female and silenced in a mated female (*Feng et al., 2014*; *Wang et al., 2020*). To precisely manipulate SAGs, we generated a new split-GAL4 line (SAG-SS3; *Figure 3F*) as SAG-SS1 labels off-target neurons that have arborizations in the SEZ that could impact feeding behavior and SAG-SS2 expression is weak making neither tool ideal (*Feng et al., 2014*). Next, we acutely silenced SAGs by driving the expression of the green-light gated anion channelrhodopsin (GtACR1) with SAG-SS3. Monitoring consumption over time, we found that acutely silencing SAGs significantly increased sucrose consumption (*Figure 3G* and *Figure 3—figure supplement 2A*). Although an off-target ascending neuron is also labeled in SAG-SS3, referred to as SAGb, it is not responsible for changes in sucrose consumption (*Figure 3—figure supplement 3*), arguing that the increased sucrose consumption observed with SAG-SS3 is due to the activity of SAG neuron itself. Downstream of SAG, we found that acutely silencing pC1 also significantly increased sucrose consumption (*Figure 3H and I*, and *Figure 3—figure supplement 2B–D*), demonstrating that pC1 neurons regulate postmating sucrose feeding. Taken together, we conclude that the increased sugar consumption after mating is part of the repertoire of behavioral changes induced after copulation by SP acting on the SPSN-SAG-pC1 mating status circuit.

# Descending neurons that regulate egg laying and sexual receptivity do not influence sucrose consumption

Postsynaptic to pC1, the mating status circuit splits into separate neural pathways that mediate different behavioral subprograms, including circuits that elicit changes in sexual receptivity (*Wang et al., 2021*) or egg-laying (*Wang et al., 2020*). We hypothesized that the neural circuit supporting the postmated increase in sucrose consumption could either follow one of the established circuits or

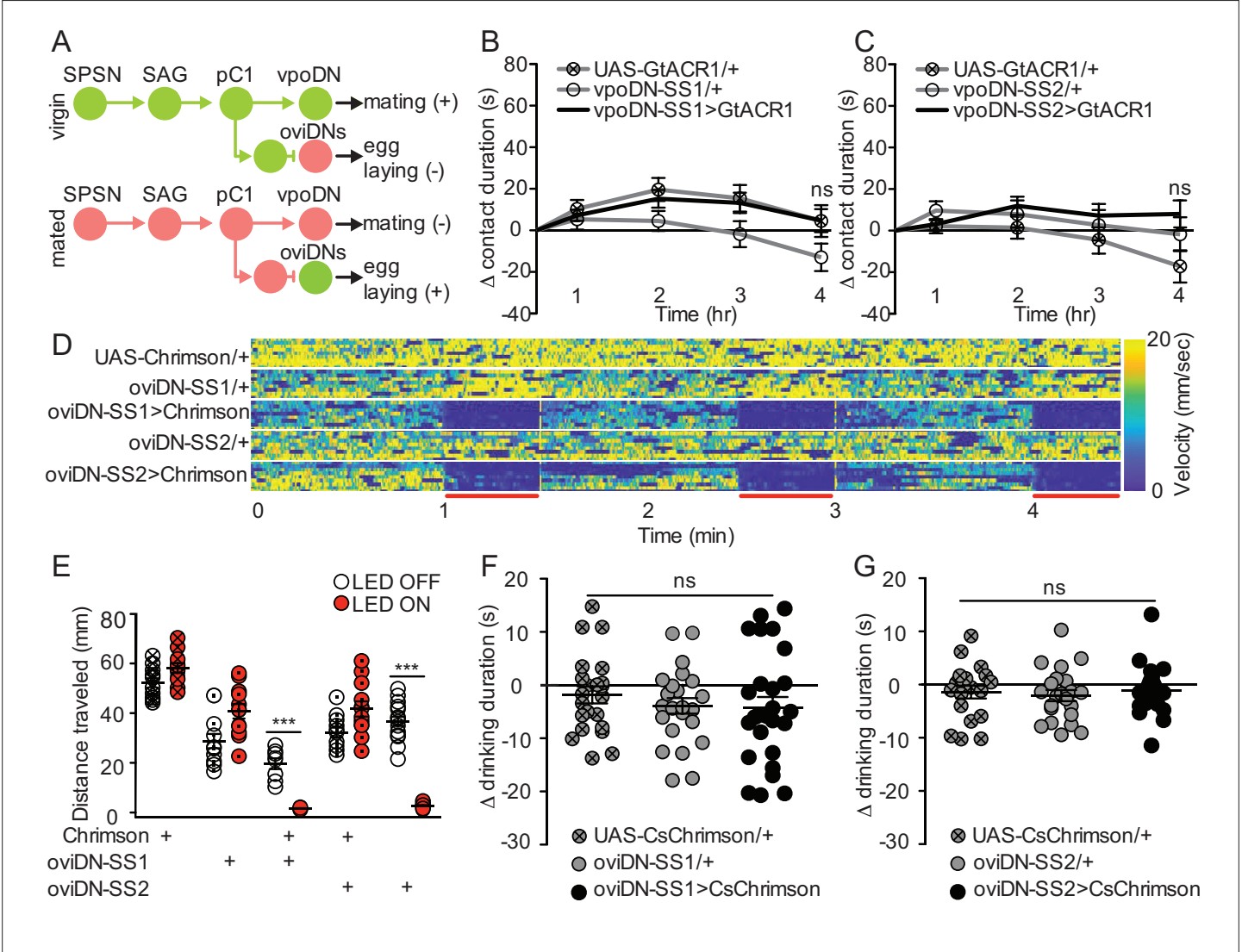

**Figure 4.** Descending neurons that regulate egg laying and sexual receptivity do not influence sucrose consumption. (**A**) Schematic of the mating circuit with neural outputs of pC1 and resulting postmating response. Green circles indicate active neurons, salmon circles indicate silenced neurons. (**B and C**) Difference in drinking time of 250 mM sucrose of indicated genotype in fly liquid interaction counter (FLIC) over 4 hr. Each data point represents the normalized drinking time of a single female in light conditions (drinking time minus the daily average drinking time of all females in dark conditions within a genotype, n=20–24), Kruskal-Wallis. Line graph shows mean and s.e.m. over time. (**D**) Velocity heatmap upon transient activation in individual walking flies, genotype indicated on the left. Each row represents the velocity of a single fly. Red bars indicate light ON (n=8–10). (**E**) Total distance traveled in the locomotor assay in (**D**). ON condition (red) includes 3–30 s light ON conditions. OFF condition (white) includes 3–30 s light OFF conditions immediately preceding each ON condition, Mann-Whitney, ***p<0.001. Aligned dot plot shows mean +/− s.e.m. (**F and G**) Drinking time of 250 mM sucrose of females exposed to light normalized to dark controls of indicated genotype in temporal consumption assay (n=19–26), One-way ANOVA, Bonferroni's post hoc. ns = not significant. Scatter plot shows mean +/− s.e.m. See *Supplementary file 1* for full genotypes. See also *Figure 4—figure supplement 1*.

The online version of this article includes the following figure supplement(s) for figure 4:

**Figure supplement 1.** Drinking times of virgin females upon optogenetic manipulation of mating status circuit output circuit neurons: vpo and oviDNs.

diverge after pC1. To test if sucrose feeding is executed by previously identified postmating subcircuits, we optogenetically manipulated the activity of the descending neurons (DNs) for sexual receptivity (vpoDNs) or egg laying (oviDNs) to mimic the mated state (*Figure 4A*) and measured sucrose consumption. We found no difference in sucrose consumption upon manipulation of vpoDNs or oviDNs (*Figure 4B, C, F and G*, and *Figure 4—figure supplement 1*). We note that we measured sucrose intake in immobilized flies for oviDN manipulations, as activating oviDNs caused females to stop walking (*Figure 4D and E*), likely a natural component of egg laying. These studies argue that neither of the previously identified circuits responsible for a single postmating response, egg laying, and sexual receptivity, mediate postmating sucrose feeding.

## pCd-2 neural outputs of SAG and pC1 mediate postmating sucrose consumption

To identify novel pathways that mediate changes in mated feeding behavior, we examined other major outputs of the mating status circuit. We identified postsynaptic partners of SAG and pC1 using automated analysis (*Buhmann et al., 2021*; *Heinrich et al., 2018*) of a fly brain electron microscopy (EM) connectome (*Zheng et al., 2018*; *Dorkenwald et al., 2022*). This approach identified three pCd-2 neurons (*Kimura et al., 2015*; *Nojima et al., 2021*), pCd-2a, pCd-2b, and pCd-2c (*Figure 5A*), that are strongly connected to SAG (39 synapses) and pC1 (146 synapses). These neurons are promising candidates to mediate postmating feeding changes because their processes descend into the prow region of the Subesophageal Zone (SEZ; *Figure 5B and C*, and *Figure 5—figure supplement 1A*), a brain region implicated in energy homeostasis and feeding (*Shiu et al., 2022*). As pC1 and SAG are cholinergic (*Feng et al., 2014*; *Wang et al., 2020*), pCd-2 cells likely receive excitatory signals from these two upstream synaptic partners and, therefore, would be predicted to be less active in mated females (*Figure 5D*).

We generated split-GAL4 lines to genetically access pCd-2a and pCd-2b (*Figure 5E and F*) and found that both neurons are female-specific (*Figure 5—figure supplement 1B*), consistent with a role in female postmating behavior. To test if pCd-2a and pCd-2b are functionally connected with pC1, we optogenetically activated pC1 using R40F04-LexA (see Methods and *Figure 5—figure supplement 2*) while simultaneously measuring calcium responses in either pCd-2a or pCd-2b neurons. In both cases, neural activation caused significant calcium increases in pCd-2a and pCd-2b neurons (*Figure 5G and H*), validating pCd-2a and pCd-2b as targets of the mating status pathway. When we optogenetically silenced pCd-2a or pCd-2b neurons in virgins, mimicking the mated state (*Figure 5D*), females significantly increased sucrose consumption compared to controls (*Figure 5I and J*, and *Figure 5—figure supplement 3C–J*). Importantly, silencing pCd-2a or pCd-2b did not influence egg laying (*Figure 5K* and *Figure 5—figure supplement 3A–B*), demonstrating the separation of circuits for increased sucrose consumption from other behaviors downstream of pC1.

To explore if optogenetic manipulation of pCd-2a or pCd-2b mimicked the mated state, we silenced these neurons in already mated females. No difference in sucrose consumption would suggest that artificial silencing imitates the mated firing pattern. Instead, we found that mated females with pCd-2a or pCd-2b silenced further increased sucrose consumption (*Figure 5—figure supplement 4*), suggesting that optogenetic inhibition is stronger than mating-induced inhibition. This is reasonable since the mated state likely decreases the activity of the mating status circuit rather than silences it completely (*Feng et al., 2014*). Next, we activated pCd-2a or pCd-2b in a mated female to mimic the virgin state to test if this cell type is necessary for the postmated increase in sugar intake. We found that mated females with pCd-2a artificially activated significantly reduced sugar intake compared to controls, recapitulating the virgin phenotype in mated females (*Figure 5—figure supplement 4*). However, mated females with pCd-2b artificially activated did not differ from controls (*Figure 5—figure supplement 4*), which may be a reflection of the fewer number of synapses from SAG and pC1 (*Figure 5A*). Thus, pCd-2a and pCd-2b activity regulate sucrose consumption in virgin and mated females, with decreased activity promoting consumption after mating.

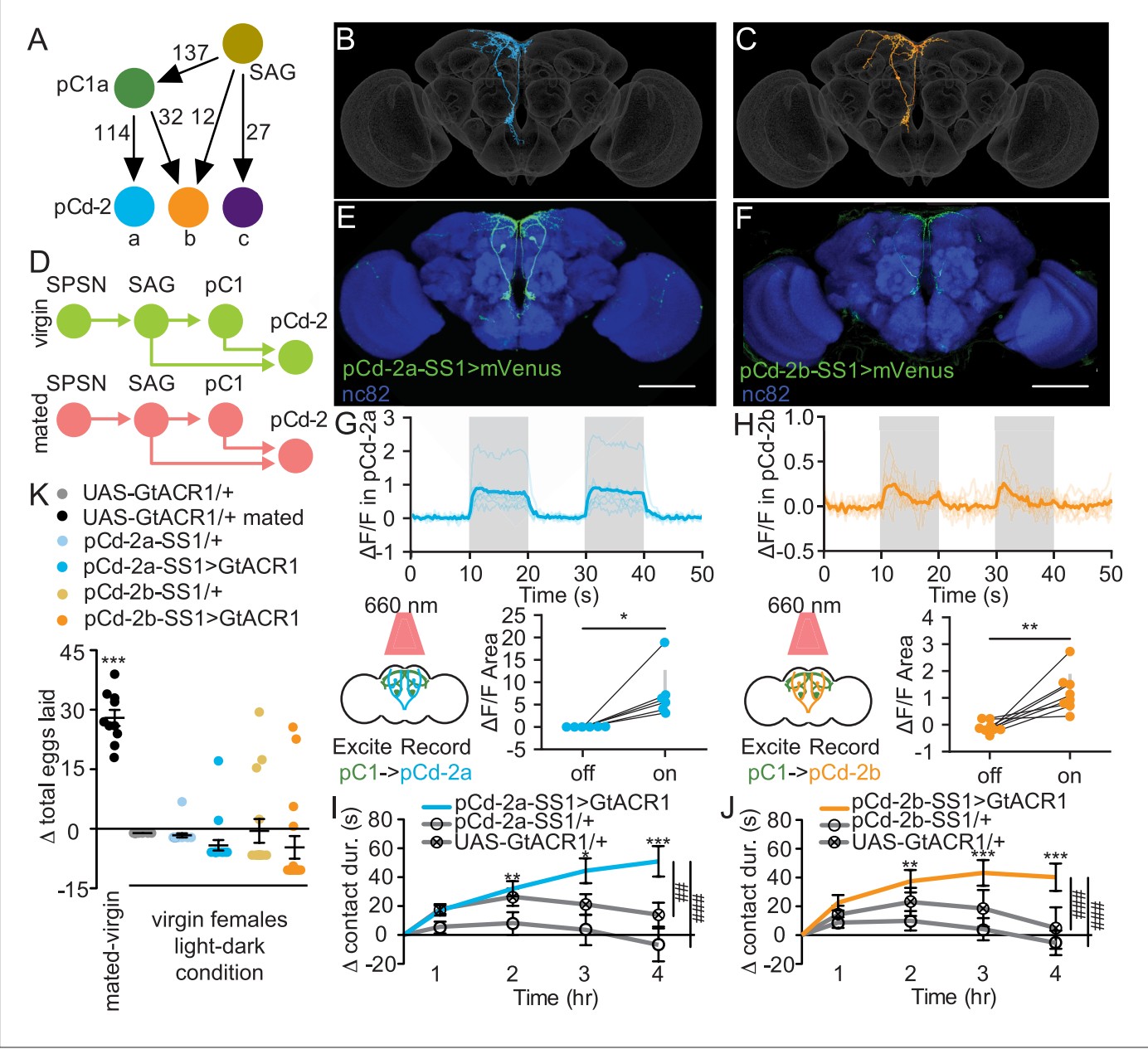

**Figure 5.** pCd-2 neural outputs of sex peptide abdominal ganglia (SAG) and pC1 mediate postmating sucrose consumption. (**A**) Schematic of neural connectivity of the mating status circuit and pCd-2 cells. Arrows represent direct connection and the number indicates the number of synapses per hemisphere. (**B and C**) Electron microscopy reconstruction of pCd-2a (**B**) and pCd-2b (**C**) neurons. (**D**) Model of circuit activity in virgin and mated state. Green circles indicate active neurons, salmon circles indicate silenced neurons. (**E and F**) Confocal images of pCd-2a-SS1 and pCd-2b-SS1 brains, stained to reveal pCd-2 projection pattern (UAS-GFP, green) and all synapses (nc82, blue). Scale bar 50 μm. (**G and H**) Calcium responses of pCd-2a (**G**) and pCd-2b (**H**) in the prow region of the brain to activation of upstream pC1 neurons with R40F04-LexA (n=6–8) and the analysis of area under the curve, Wilcoxon Rank Test, scatter plot shows mean +/− s.e.m. (gray bar). Gray shading represents optogenetic stimulation. (**I and J**) Drinking time of 250 mM sucrose of females exposed to light normalized to dark controls of indicated genotype in fly liquid interaction counter (FLIC) over 4 hr (n=22–26). Line graph shows mean and s.e.m. over time, Kruskal-Wallis, Dunn's post hoc at 4 hr, ##p<0.01, ###p<0.001. (**K**) Difference in eggs laid in 24 hr of females exposed to light normalized to dark controls (n=10–20). For 'UAS-GtACR1 mated' group, the dataset represents eggs laid by mated females normalized to virgin females of the same genotype. Scatter plot shows mean +/− s.e.m, one sample t-test. *p<0.05, **p<0.01, ***p<0.001. See *Supplementary file 1* for full genotypes. See also *Figure 5—figure supplement 1*, *Figure 5—figure supplement 2*, *Figure 5—figure supplement 3*, and *Figure 5—figure supplement 4*.

The online version of this article includes the following figure supplement(s) for figure 5:

**Figure supplement 1.** Anatomical characterization of pCd-2 split-GAL4 lines.

**Figure supplement 2.** R40F04-LexA labels pC1.

**Figure supplement 3.** Independent pCd-2a and pCd-2b lines regulate postmated sucrose consumption.

**Figure supplement 4.** pCd-2 cell type is both necessary and sufficient to drive postmated sucrose intake.

## Sexually dimorphic neuroendocrine Lgr3+ cells receive mating status signals via pCd-2 neurons and regulate sucrose consumption in mated females

To examine how pCd-2 neurons influence sucrose consumption, we used EM analyses and identified the major downstream targets of pCd-2 as t-shape cells (872 synapses) having processes in the

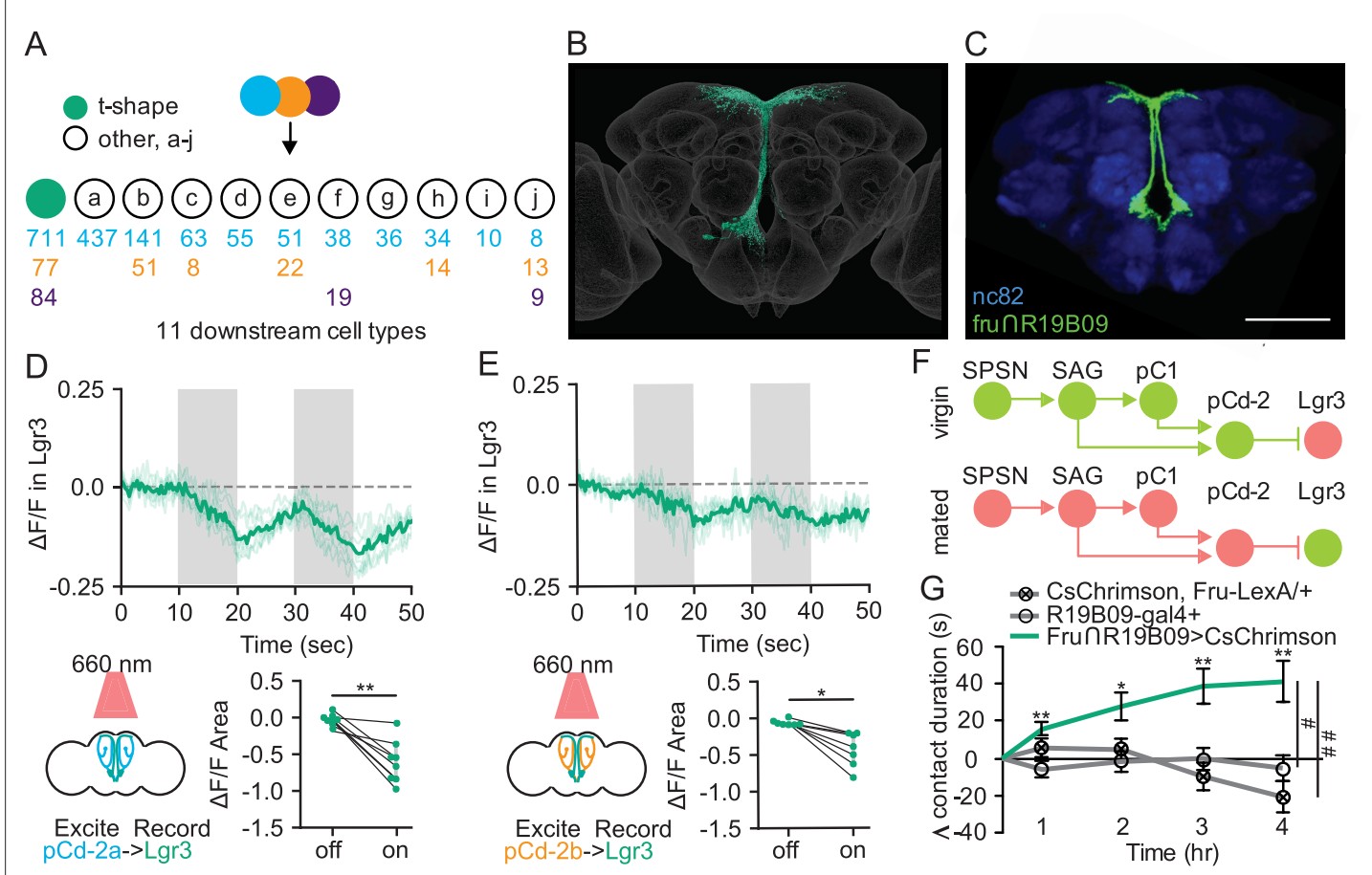

**Figure 6.** Sexually dimorphic neuroendocrine Lgr3+ cells receive mating status signals via pCd-2 neurons and regulate sucrose consumption in mated females. (**A**) Downstream neuronal cell types of pCd-2a (blue), pCd-2b (orange), and pCd-2c (purple) cells identified by electron microscopy reconstruction. Arrow indicates direct connection and numbers indicate pCd-2 output synapses with a color corresponding to upstream pCd-2 cell (a, b, or c). T-shape cell type is represented with a turquoise circle; other cell types are represented with white circles (a-j). (**B**) Electron microscopy reconstruction of t-shape neurons. (**C**) Confocal image of fru+ Lgr3-expressing cells (see methods) in the medial brain (UAS-CD8:GFP, green) and all synapses (nc82, blue). Scale bar 50 μm. (**D and E**) Calcium responses of Lgr3-expressing cell bodies in the prow region of the brain to activation of upstream pCd-2a (**D**) or pCd-2b (**E**) neurons (n=7–8) and the analysis of area under the curve, Wilcoxon Rank Test, scatter plot show mean +/− s.e.m. (gray bar). Gray shading represents optogenetic stimulation. (**F**) Neural model for the coordination of mating status and sucrose intake. Green circles indicate active neurons, salmon circles indicate silenced neurons. (**G**) Drinking time of 250 mM sucrose of females exposed to light normalized to dark controls of indicated genotype in fly liquid interaction counter (FLIC) over 4 hr (n=10–15). Line graph shows mean and s.e.m. over time, Kruskal-Wallis, Dunn's post hoc at 4 hr, #p<0.05, ##<0.01. *p<0.05, **p<0.01, ***p<0.001. See ***Supplementary file 1*** for full genotypes. See also ***Figure 6—figure supplement 1***.

The online version of this article includes the following figure supplement(s) for figure 6:

**Figure supplement 1.** The main output cell type of pCd-2 neurons are t-shape Lgr3 neurons.

SEZ prow region and a neuroendocrine center, the pars intercerebralis, with terminal arbors similar to insulin-producing cells and other median secretory cells (*Liao and Nässel, 2020*; *Figure 6A* and *Figure 6—figure supplement 1*). Based on anatomical similarity, these cells are leucine-rich repeat G-protein-coupled receptor-expressing medial bundle neurons (Lgr3 + mBNs; *Dolan et al., 2007*; *Figure 6B and C*). Lgr3 expression is sexually dimorphic, with a greater number of cells in the central brain and abdominal ganglia of females (*Meissner et al., 2016*), consistent with a role in female behavior. Moreover, Lgr3 is a receptor for the *Drosophila* insulin-like peptide 8 (Dilp8; *Colombani et al., 2015*; *Garelli et al., 2015*; *Vallejo et al., 2015*), and both Lgr3 and Dilp8 are known to regulate feeding (*Yeom et al., 2021*). These findings suggest that the Lgr3+ mBNs may represent the site where mating status circuits interact with hunger/satiety circuits to regulate feeding.

To test this hypothesis, we first examined whether Lgr3+ mBNs are functionally connected to pCd-2a or pCd-2b by optogenetically activating these populations while simultaneously measuring calcium responses in Lgr3+ neurons. In both cases, neural activation caused significant calcium decreases in Lgr3+ neurons (*Figure 6D and E*), validating Lgr3+ mBNs as targets of pCd-2 and the mating status pathway. As pC1 is inhibited in the mated state, our studies argue that Lgr3+ mBNs are more active in mated females (*Figure 6F*). To examine whether activity in Lgr3+ mBNs influences consumption, we targeted this population with a genetic intersectional approach (R19B09∩fru; *Figure 6C*; *Meissner et al., 2016*), optogenetically activated Lgr3 + mBNs, and monitored sucrose consumption. We found that increased activity in Lgr3+ mBNs promotes sucrose consumption, mimicking the mated state (*Figure 6G*). Together, these studies reveal that Lgr3+ mBNs receive inputs from the canonical postmating circuit and are part of a neuroendocrine pathway that regulates sucrose consumption in postmated females.

## Discussion

Animals must pair their internal metabolic needs with external food choices to maintain homeostasis. As animals' nutritional requirements shift in a state-dependent manner, they must adapt their feeding behavior accordingly, necessitating mechanisms that couple dynamic physiological demands with nutritional consumption. After mating, females experience a new metabolic requirement related to increased offspring production. In *Drosophila*, like many insect species, this is demonstrated through the significant increase in egg production and egg laying, two energetically expensive tasks. We predicted that mated females require more calories than virgins and, therefore, exhibit increased consumption of sugar, their main energy source (*Simpson et al., 2015*).

Here, we characterized the sucrose consumption of virgin and mated females and found that mated females demonstrate increased sucrose intake. Although mated females are predicted to need more sugar to support increased reproductive output, we find that the mechanism that regulates postmating sugar intake is independent of the caloric deficiencies caused by egg production and is anticipatory in nature. This allows females to increase caloric intake prior to experiencing a deficiency in internal sugar levels. The postmated increase in sugar feeding is due primarily to an elongation of feeding bouts rather than an increased likelihood of initiating a feeding event or an increase in the appetitive nature of sugar. This is supported by data that demonstrates no change in response of the sucrose-sensing gustatory sensory neurons across mating states. Instead, we find that the female's mating status impinges on an endocrine center of the brain, where it may associate the mated state with elevated hunger levels.

In virgins, elevated activity of SAG and pC1 potentiates pCd-2-mediated inhibition of Lgr3+ cells, resulting in low sugar intake. After mating, the activity of SAG and pC1 is reduced (*Feng et al., 2014*; *Wang et al., 2020*), decreasing the activity of pCd-2 and alleviating the inhibition from *Lgr3+* cells, increasing sucrose intake. Lgr3 is a leucine-rich repeat-containing G-protein coupled receptor, which binds Dilp8 (*Drosophila* insulin-like peptide 8; *Colombani et al., 2015*; *Garelli et al., 2015*; *Vallejo et al., 2015*) to regulate feeding in flies. Expression levels of Lgr3 and Dilp8 rise in fed flies and mutations in either gene increase feeding, arguing that Dilp8 and Lgr3 are satiety factors (*Yeom et al., 2021*). Our calcium imaging and behavioral studies of Lgr3+ mBNs indicate that they are more active in a mated state and activation of these neurons promotes sucrose consumption. Importantly, it has not been established how Dilp8 influences the activity of Lgr3+ mBNs. Regardless, together these data suggest a proposed model for how Lgr3+ mBN neurons integrate signals from the mating status and hunger/satiety systems (*Figure 7*). In virgin or satiated female flies, pCd-2 activity and circulating

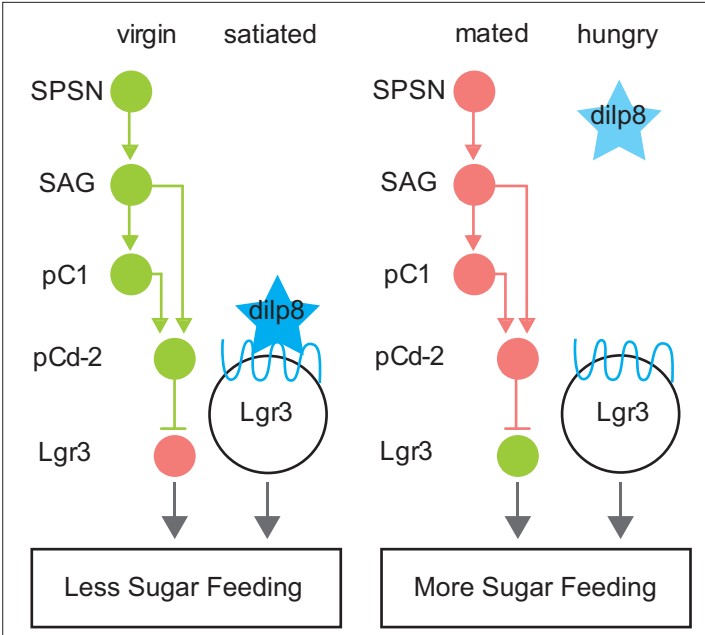

**Figure 7.** Model summarizing neuroendocrine Lgr3-dependent feeding modulation. Schematic of neural connectivity of mating status circuit (SPSN-SAG-pC1), mating status output neurons (pCd-2), and downstream Lgr3 cells. Circles represent neurons of indicated cell type. Lines between neurons represent synaptic connectivity: arrowheads indicate excitatory synapses, and horizontal lines indicate inhibitory synapses. Circuitry shown in green is active, pink indicates silenced. Star (blue) represents Dilp8. Blue wavey line is Dilp8 receptor, Lgr3. Status is indicated at the top and feeding outcome is indicated at the bottom (boxed).

DILP8, respectively, inhibit Lgr3+ mBNs, reducing sugar feeding. Conversely, in mated or hungry flies, Lgr3+ mBNs lack the inhibitory signal from pCd-2 neurons or circulating Dilp8, respectively, resulting in increased sugar consumption. Hence, the mated state may be viewed mechanistically as an additional 'hunger' signal within this system.

Consistent with the neural regulation of other postmating behaviors, including other postmated changes in feeding, the three-tier mating status circuit (SPSN-SAG-pC1) regulates postmated changes in sucrose consumption. Just as the silencing of first-order SPSNs and second-order SAG neurons drive changes in salt and protein appetites (*Ribeiro and Dickson, 2010*; *Walker et al., 2015*), we also find that artificially silencing any part of the SPSN-SAG-pC1 circuit drives increased sugar feeding, recapitulating the mated phenotype. Moreover, the neural regulation of postmating sucrose consumption, like other postmated behaviors, is also regulated by an independent pathway postsynaptic to the mating status circuit. Although the three-tier circuit appears to be a master regulator, the network splits into dedicated neural pathways modulating specific postmating behaviors such as egg laying (*Wang et al., 2020*), sexual receptivity (*Wang et al., 2021*), or sucrose consumption. Interestingly, although increased sugar intake may indeed support egg production, the neural mechanisms regulating the two behaviors are independent downstream of pC1.

Our findings provide insight into not only the mechanisms that couple dynamic state-dependent physiological demands with nutritional consumption but also how these mechanisms are distinguishable from food deprivation-induced nutrient need. Sugar and yeast are both vital nutrients for flies (*Simpson et al., 2015*). Female *Drosophila* increase feeding of both sugar and yeast after nutrient deprivation and after mating (*Ribeiro and Dickson, 2010*; *Corrales-Carvajal et al., 2016*). Upon nutrient deprivation, the sensitivity of gustatory neurons change, promoting the detection of the deprived nutrient, for both sugar and yeast (*Inagaki et al., 2012*; *Steck et al., 2018*). However, we found that mated-related changes in sucrose were not due to changes in sensory neuron sensitivity. Similarly, mated-related changes in gustatory neuron sensitivity to yeast have not been found (*Steck et al., 2018*). It should be noted that after mating females display a transient modulation in chemosensory neurons that respond to polyamines (*Hussain et al., 2016*). However, these changes are not associated with polyamine consumption but instead oviposition behavior (*Hussain et al., 2016*).

Taken together, these studies suggest that changes in postmated feeding behavior result from the mating status circuit (or its output neurons) modulating nutrient-sensing or feeding circuits, rather than directly tuning sensory neurons to promote feeding.

Although mating status impinges on circuits that either process nutrient-sensing signals or regulate feeding circuits, there is no shared circuit yet identified for elevated nutrient intake of sugar and yeast in mated females. Beyond the three-tier mating status circuit, no overlap between areas of the brain modulating postmated increases in protein (*Liu et al., 2017*; *Münch et al., 2022*) and sugar consumption (this manuscript) has been found. It remains to be elucidated whether the circuit described here specifically modulates post-mating changes in sucrose consumption or whether it impacts the intake of other nutrients, including protein and salt.

Overall, our findings show that an elevated appetite for sucrose is an important behavioral subprogram elicited by mating and executed by female-specific circuitry, shifting the physiology of a mated female. The activation of hunger centers by the mating status circuit provides a neural mechanism that anticipates the large energetic demands associated with offspring production and increases caloric intake, promoting reproductive success.

## Materials and methods

### Rearing conditions and strains

Flies were reared on standard cornmeal-agar-molasses medium, at 25 °C, 65% humidity on a 12-hr-light-dark cycle. Flies used in optogenetic assays were reared on food containing 0.25 mM all-trans-retinal (Sigma-Aldrich) in darkness, before and after eclosion.

Flies were collected under $CO_2$ 1–8 hr after eclosion and housed in same-sex groups. To generate mated females, 15 virgin females, and five males were paired 3–5 days post-eclosion and group-housed for 72 hr. To generate recently mated females, a single virgin female was paired with a single male for 2 hr. If mating was observed, the recently mated female was then immediately placed into the indicated assay. To generate 24 hr mated females, a single virgin female was paired with a single male for 2 hr. If mating was observed, the recently mated female was then group housed with other recently mated females until testing. To generate egg-less females we used either ovoD mutant females (*Mével-Ninio et al., 1996*) or hsp70-bam females (*Ohlstein and McKearin, 1997*) following the heat-shock protocol (*Walker et al., 2015*).

### Capillary feeding (CAFE) assay

Capillary feeding arenas were modified from the original (*Ja et al., 2007*). Briefly, we equipped a plastic vial, containing a wet kimwipe, with an altered rubber stopper lid housing two truncated 200 ml pipette tips whereby the capillaries can be inserted through. The capillaries (calibrated glass micropipettes, 5 ml) were loaded with 50 mM sucrose solution containing 0.25 mg/ml blue dye. To measure sucrose consumption, five virgin or mated females were gently aspirated into an arena for 24 hr, the displacement of the meniscus was measured, and the volume per fly was calculated (1 mm = 0.038 ml). Experiments were completed on at least three different days with all groups tested on each day.

### Fly liquid interaction counter (FLIC)

Females of indicated genotype or mating status were first wet-starved for 12–18 hr. Following this, females were then gently aspirated into the fly liquid food interaction counter (FLIC; *Ro et al., 2014*), which was pre-filled with a sucrose solution of indicated concentration. The voltage from each well was continuously recorded and a .csv file was produced. Significant changes in the voltage indicated that the fly was contacting the liquid. Contact duration acted as a proxy for consumption time. Based on a video analysis of flies drinking in FLIC in our lab, we modified the parameters in R to better model consumption. The new parameters are as follows: Feeding.Interval.Minimum=10, Feeding.Threshold. Value=10, Tasting.Threshold.Interval=c(01,02), Feeding.Minevents=6, Signal.Threshold=10. Experiments were completed on at least three different days with all groups tested on each day.

For optogenetic control of neural activity, while flies were in the FLIC, we designed and constructed a 12-chamber lid equipped with either three red LEDs (617 nm, Luxeon Star, catalog number SP-01-E6) or three green LEDs (530 nm, Luxeon Star, catalog number SP-01-G4). The 12-chamber cover

was made from half of a 24-well plate, which was trimmed to fit. Lights were attached to half of the 24-well plate lid and fitted atop the chamber cover. Lights were switched on (constant light) once all flies were loaded into the wells and remained on for the entire length of the assay. In the case of neural activation to observe potential decreases in sucrose consumption, flies were placed in humidified red light LED box (20 cm × 10 cm × 30 cm, custommade light box equipped with a 150 LED ribbon programmed with an Arduino with a constant light on) or a humidified dark box at the start of starvation. Data were prepared for analysis using using FLIC_automated_group_assignment (a customized R program, *Laturney, 2023*) and R (The R Project for Statistical Computing, https://www.r-project.org/).

## Proboscis extension response (PER) assay

Females of indicated mating status were either wet-starved for 12–18 hr (starved) or kept on standard food (fed). Flies were then anesthetized using $CO_2$ and fixed to a glass slide with No More Nails polish and then allowed to recover for 2 hr in a humidified box. Immediately before testing, flies were given water until they no longer responded to five consecutive presentations. Flies were then presented three times with sucrose of an indicated concentration and proboscis extension was recorded. Genotype of the fly was coded and, therefore, not apparent at the time of testing. Experiments were completed on multiple days with all groups included each day.

## Temporal consumption assay (TCA)

Flies were starved for 18 hr in darkness, anesthetized using $CO_2$, and then fixed to a glass slide with No More Nails polish with limited light and allowed to recover for 2 hr in a humidified red light LED box (20 cm × 10 cm × 30 cm, custom made light box equipped with a 150 LED ribbon programmed with an Arduino with a constant light on) or a humidified dark box. Immediately before testing, flies were given water until they no longer responded to five consecutive presentations. In testing, flies were presented with 250 mM sucrose, and consumption time was recorded. Each trial ended when the fly was presented with the tastant and no longer responded to five consecutive presentations. Flies in the dark condition were given water and tested in low light conditions. Flies in red light condition were given water and tested in the presence of constant red light (LED panel made comprised of three red LEDs: 617 nm, Luxeon Star, catalog number SP-01-E6). Genotype of the fly was coded and, therefore, not apparent at the time of testing. Experiments were completed on at least three different days with all groups included each day.

## Egg laying assay

7-day-old females were placed in a vial with standard food and placed into a custom-made light box (20 cm × 10 cm × 30 cm) that was equipped with a 150 LED ribbon programmed with an Arduino. In neural activation tests, the red light was transiently on (1 s on/1 s off) to limit the likelihood of reduced neural firing with long light exposure time. In neural silencing tests, the green light was constantly on. Half of the females of each genotype were placed in vials wrapped in foil to function as no-light condition controls. After 24 hr, females were removed and eggs were immediately counted. Genotype of the fly was coded and, therefore, not apparent at the time of egg counting. Experiments were completed on at least two different days.

## Locomotor assay

Locomotor assays were conducted as described (*Bidaye et al., 2020*). Briefly, the behavior arenas were created by pouring 1.5% agarose gel into a 150 mm Petri plate, which was then loaded with 3D-printed acrylic molds to generate four bowl-shaped depressions. Once cooled, the agarose-filled Petri-plate containing four arenas (44 mm in diameter) was topped with a glass plate painted with Sigmacote. Flies were recorded with a FLIR Blackfly S camera (FL3-U3-13Y3M-C) fitted with a focus lens (LMVZ990-IR) and MIR bandpass filter (Midopt BP850) to allow infrared imaging at a resolution of 1280 × 1024 at 30 fps. Arenas were illuminated by a custom LED plate capable of 870 nm (infrared) and 630 nm (red) wavelengths (see *Bidaye et al., 2020* for details). The 870 nm light was on throughout the assay. For CsChrimson activation, 630 nm LED was transiently switched on and pulsed at 50 Hz (performed in a dark room).

## In vivo calcium imaging with taste stimulation

Protocol was adapted (*Shiu et al., 2022*).Females aged 14–20 days: virgins were kept in groups of females and mated females were co-housed with Canton-S males. Females were food-deprived for

18–24 hr prior to imaging. A window was made in the head of the fly to allow for visualization of the SEZ and a drop of ~250 mOsmo AHL solution (*Wang et al., 2003*) was added to the head and imaging was immediately performed on a fixed-stage 3i spinning disk confocal microscope with a piezo drive and a 20 x water objective (1.6 x optical zoom). A 250 mM sucrose solution was delivered to the proboscis using a glass capillary. To analyze these images, a maximum intensity projection encompassing the arbors of Gr64f neurons was made using Fiji. Large ROIs were drawn manually corresponding to the GCaMP expression and a large ROI was drawn in an adjacent region to measure background autofluorescence. Mean fluorescence levels from the background ROI were subtracted from the Gr64f ROIs at each time point, resulting in the fluorescence trace over time: F(t). delta F/F was measured as follows: (F(t) – F(0)) / F(0), where F(0) was the average of 10-time points before stimulation with sucrose and F(t) was the average background subtracted increases in fluorescence during sucrose stimulation. Area under the curve for 10 frames before stimulation and during stimulation was approximated with the trapezoidal rule in Python using the NumPy.trapz function. Statistical analysis was carried out in Python using the SciPy package, version 1.7.3 (*Virtanen et al., 2020*). deltaF/F0 images were created as described (*Yao and Scott, 2022*). Experiments were completed over 4 days.

## In vivo functional connectivity experiments

In vivo sample preparation for calcium imaging was performed as described (*Shiu et al., 2022*). Prior to imaging, female flies were aged for 2 weeks and then food-deprived in a vial containing a distilled water-soaked kimwipe for 18–24 hr prior to imaging. The pC1a-LexA line (R40F04-LexA attP40) was identified with the use of NeuPRINT and NeuronBridge. Within NeuPRINT we queried pC1a to determine the body ID (5813046951) and then using NeuronBridge we searched for lines containing this cell type. We chose R40F04 (score: 933) due to the lack of coexpression within the prow, the preferred imaging area of pCd-2. We verified that R40F04-LexA labels pC1 and describes the expression pattern (see *Figure 5—figure supplement 2*). Female flies were housed with Canton-S males throughout aging and food deprivation. Following dissection, samples were bathed in ~250 mOsmo AHL solution (https://www.sciencedirect.com/science/article/pii/S0092867403000047) and imaged immediately using a 2-photon microscope. Volumetric images of the prow region were acquired with 920 nm light at 2.9 Hz with resonant scanning and a piezo-driven objective. During imaging, a custom ScanImage plugin was used to deliver two 10 s pulses of 660 nm light through the objective at a 10 s interval to excite CsChrimson.

Image and statistical analysis of functional connectivity experiments were performed using Fiji (https://www.nature.com/articles/nmeth.2019), CircuitCatcher (a customized Python program, *Bushey, 2019*), and Python (Python Software Foundation, https://www.python.org/). Using Fiji, image stacks for each time point were the first maximum intensity projected. Then, using CircuitCatcher, an ROI containing Lrg3+ mBN cell bodies and a background ROI of brain tissue were selected and the average fluorescence intensity for each ROI at each timepoint was retrieved. Then, in Python, background subtraction was carried out for each timepoint ($F_t$) and initial fluorescence intensity ($F_{initial}$) was calculated as the mean corrected average fluorescence intensity for frames 1–15. Then, $\Delta F/F$ was calculated using the following formula: $F_t$-$F_{initial}$/$F_{initial}$. Area under the curve for before (off; frames 1–29) and during (on; frames 30–58) light stimulation was approximated for flies that expressed both the optogenetic neural activator (Chrimson) and the calcium dependent indicator (GCaMP) as well as for the controls (flies only expressing the calcium indicator) with the trapezoidal rule in Python using the NumPy.trapz function. Traces were then normalized to controls. Statistical analysis of functional connectivity experiments was carried out in Python using the SciPy package, version 1.7.3 (https://www.nature.com/articles/s41592-019-0686-2).

## Dissections and Immunohistochemistry

All CNS dissections and immunostaining (unless directly addressed) were performed following the detailed instructions found at https://www.janelia.org/project-team/flylight/protocols. To image split-GAL4 lines and intersections, we used the following primary antibodies: mouse α-Brp (nc82, DSHB, University of Iowa, USA) at 1:40, chicken α-GFP (Invitrogen, A10262) at 1:1000; and the following secondary antibodies: goat α-mouse AF647 (Invitrogen, A21236) at 1:500 and goat α-chicken AF488 (Life Technologies, A11039) at 1:500. Multi-Color Flip-Out fly generation was performed following the protocol (*Nern et al., 2015*). For imaging, we used the following primary antibodies: mouse α-Brp

(nc82, DSHB, Univerisity of Iowa, USA) at 1:40, rabbit α-HA (rabbit α-HA, Cell Signaling Technology, C29F4) at 1:1000, rat α-Flag (Anti-DYKDDDDK, Novus Biologicals, NBP1-06712) at 1:1000; and the following secondary antibodies: goat α-mouse AF647 (Invitrogen, A21236) at 1:500, goat α-rabbit AF488 (Invitrogen, A11034) at 1:500, and goat α-rat AF568 (Invitrogen, A11077) at 1:500. To colabel R40F04-LexA and pC1, we used the following primary antibodies: mouse α-Brp (nc82, DSHB, University of Iowa, USA) at 1:40, chicken α-GFP (Invitrogen, A10262) at 1:1000 and rabbit α-DsRed (Living Colors, 632496) at 1:1000; and the following secondary antibodies: goat α-mouse AF647 (Invitrogen, A21236) at 1:500, goat α-chicken AF488 (Life Technologies, A11039) at 1:500, goat α-rabbit AF568 (Invitrogen, A11011) at 1:500.

Reproductive tract dissections and staining were performed as described (*Billeter and Goodwin, 2004*). For the primary antibody, we used chicken α-GFP (Invitrogen, A10262) at 1:1000 followed by goat α-chicken AF488 (Life Technologies, A11039) at 1:500 with the addition of Alexa Fluor 633 phalloidin (Thermo Fisher Scientific) to visualize the muscle-lined reproductive tract (1:500).

## Confocal imaging

Samples were imaged on an LSM710 confocal microscope (Zeiss) with a Plan-Apochromate 20 × 0.8 M27 objective and images were prepared in Fiji.

## Electron microscopy neural reconstructions and connectivity

SAG was previously partially reconstructed (*Wang et al., 2020*) in the Full Adult Female Brain (FAFB; *Zheng et al., 2018*) electron microscopy dataset using the CATMAID software (*Saalfeld et al., 2009*). We completed additional tracing of this neuron by continuing branches with a combination of both manual and assisted tracing. Manual tracing consisted of generating a skeleton of the neuron by following the neuron and marking the center of each branch. Assisted tracing consisted of joining and proofreading pre-assembled skeleton fragments with automated segmentation (*Li et al., 2019*). In addition to the skeleton tracing, new chemical synapses were also annotated as previously described (*Zheng et al., 2018*). Finally, downstream synaptic targets of SAG were then traced out from these additional locations using both manual and assisted tracing techniques as described above.

Neurons traced in CATMAID, including SAG, pC1a, egg-laying circuit neurons (*Wang et al., 2020*), and sexual receptivity circuit neurons (*Wang et al., 2021*), were all located in Flywire (flywire.ai), which uses the same EM dataset (*Zheng et al., 2018*). To identify synaptic partners, we used a connectome annotation versioning engine (CAVE; *Buhmann et al., 2021*; *Heinrich et al., 2018*) using a cleft score cutoff of 100 to generate synapses of relatively high confidence (*Heinrich et al., 2018*).

## Intersectional method to access SPSNs

Based on previous reports, three genetic drivers label the SPSNs (VT003280, fru, and ppk; *Feng et al., 2014*; *Häsemeyer et al., 2009*; *Yang et al., 2009*) with driver lines available in two separate binary expression systems (UAS-GAL4 and LexA-LexAop). With the use of a conditional reporter line (UAS > stop > GFP) and in combination with an inducible FLP line (LexAopFLP) three combinations were produced: fru-LexA∩ppk-GAL4, fru-LexA∩VT3280-GAL4, and VT3280-LexA∩ppk-GAL4. Images of the female reproductive tract were used to confirm the labeling of the SPSNs and images of the brain to determine off-target neural expression. With the use of a conditional effector line (UAS > stop > KIR or UAS > stop > TNT), SPSNs were silenced and females (along with genetic controls) were tested for changes in sucrose consumption.

## Intersectional method to access Lgr3 median bundle neurons

We used R19B09-GAL4 to label Lgr3+ neurons. R19B09 is a fragment from the largest *Lgr3* intron and confers sexually dimorphic median bundle expression similar to that exhibited by another driver line Lgr3-GAL4::VP16. This line was made from a bacterial artificial chromosome encompassing the Lgr3 locus and the VP16 activation domain inserted in the place of the first exon of *Lgr3* (*Meissner et al., 2016*). Lgr3-GAL4::VP16 colocalizes with the Lgr3 probe, providing evidence that this driver line faithfully reveals Lgr3 expression (*Meissner et al., 2016*). Furthermore, R19B09-LexA and Lgr3-GAL4::VP16 colabel the median bundle neurons, suggesting that R19B09 is also a faithful reporter of *Lgr3* in this region (*Meissner et al., 2016*). Therefore, we assume the majority, if not all, of the cells labeled by R19B09-GAL4 express *Lgr3*. Previous reports examining the number of cells labeled by the

intersection of Lgr3-GAL4::VP16 and Fru-LexA report labeling 20.6 neurons per female brain hemi-sphere (*Meissner et al., 2016*). We assume that the intersection of R19B09-GAL4 and Fru-LexA labels a similar number of cells given that they both label the sexually dimorphic median bundle neurons (*Meissner et al., 2016*). Based on the electron microscopy connectome, we predict there are 31 't-shape' neurons downstream of pCd-2. Therefore, we predict we are gaining access to 66% of this downstream cell type using this method.

Similar to labeling the SPSNs, we used a conditional reporter line (UAS > stop > GFP), and in combination with an inducible FLP line (LexAopFLP) we produced fru-LexA∩R19B09-GAL4. Images of the female central nervous system were used to confirm the labeling of the median bundle neurons and substantiate that no off-target neurons were labeled. With the use of a conditional effector line (UAS > stop > CsChrimson or UAS > stop > KIR), the Lgr3+ MBNs were activated or silenced and females (along with genetic controls) were tested for changes in sucrose consumption.

## Split-GAL4 screening and stabilization

### pCd-2 split generation
Using the FlyEM Hemibrain V1.2.1 dataset via NeuPRINT, we queried 'SAG' and explored the output neuron list, identifying three pCd-2 neurons per hemisphere (SMP286, SMP287, SMP288). From here, we used NeuronBridge (*Meissner et al., 2022*; *Clements et al., 2022*) to manually explore the light microscopy matches and generated a list of possible hemidriver that labeled this cell type. The expression pattern of the p65ADZp and ZpGAL4DBD combinations were examined by driving the expression of the UAS reporter (20xUAS-CsChrimson-mVenus in attP18). Combinations with specific expression in pCd-2 neurons were stabilized.

### SAGb split generation:
Using NeuronBridge (*Meissner et al., 2022*; *Clements et al., 2022*), we queried 'VT050405' (the genetic construct used as the p65ADZp hemidriver in SAG-SS3) and identified multiple 'Body Ids' that morphologically matched the non-SAG ascending cell type. From here, we generated a 'Color Depth Search' to create a list of possible hemidriver that labeled this cell type. The expression pattern of the p65ADZp and ZpGAL4DBD combinations were examined by driving the expression of UAS reporter (20x UAS-CsChrimson-mVenus in attP18) and combinations with specific expression in SAGb neurons were stabilized. Expression pattern of the split-GAL4 was then compared to SAGb. We confirmed the cells match in morphology with the following characteristics: the location of the soma, ladder rung arborizations in the ventral nerve cord, bowing ascending axons in the SEZ, and arborizations in the dorsal protocerebrum. See *Figure 3—figure supplement 3*.

## Statistical analysis
Statistical tests for all experiments, with the exception of in vivo calcium imaging and in vivo functional connectivity experiments, were performed in GraphPad Prism. For two- and three-group comparisons, data were first tested for normality with the KS normality test (alpha = 0.05). If all groups passed then groups would be compared with a parametric test (t-test or one-way ANOVA, respectively). If at least one group did not pass, groups were compared with a non-parametric version (Mann-Whitney test or Kruskal-Wallis test, respectively). For all multi-factorial designs, a two-way ANOVA was performed with a Bonferroni post-hoc test. All statistical tests, significance levels, and the number of data points (N) are specified in the figure legend.

## Normalized datasets
Most datasets from optogenetic assays were normalized within each genotype. To generate this normalized dataset, data from females within the no-light condition were averaged, creating a 'no-light mean' for each genotype. This value was subtracted from each individual female within the light condition of the corresponding genotype. This dataset was then graphed and statistical analysis was performed as outlined above.

## Data availability

Detailed information about R code used to extract feeding behavior using our modified parameters for each experimental set-up is found at https://github.com/mlaturney/FLIC_automated_group_assignment, (*Laturney, 2023*).

## Acknowledgements

Members of the Scott lab provided contributions to experimental design, data analysis, and manuscript preparation. Salil Bidaye assisted with the analyses of locomotor data. This work was supported by NIH R01DC013280 (KS). Neuronal reconstruction for this project took place in a collaborative CATMAID environment in which 27 labs are participating to build connectomes for specific circuits. Therefore, work done for published (*Wang et al., 2020*) and ongoing projects has proved useful to us. Development and administration of the FAFB tracing environment and analysis tools were funded in part by National Institutes of Health BRAIN Initiative grant 1RF1MH120679-01 to Davi Bock and Greg Jefferis, with software development effort and administrative support provided by Tom Kazimiers (Kazmos GmbH) and Eric Perlman (Yikes LLC). We thank Peter Li for sharing his automatic segmentation (https://doi.org/10.1101/605634). Tracing in Cambridge was supported by Wellcome Trust (203261/Z/16/Z) and ERC (649111) awards to G Jefferis. We thank the Princeton FlyWire team and members of the Murthy and Seung labs for the development and maintenance of FlyWire (supported by BRAIN initiative grant MH117815 to Murthy and Seung). We found the partial tracing of SAG and pC1 in CATMAID useful (*Wang et al., 2020*). In addition to our tracing efforts, we are appreciative of the tracing accomplished by other labs. For help with tracing of pC1a in FlyWire, we thank the Murthy and Seung labs (77%) and the Jefferis lab (7%). For help with tracing pCd-2 cells we would like to thank the Dickson lab (CATMAID: 76% of pCd-2a, 44% of pCd-2b, 28% of pCd-2c), the Jefferis lab (CATMAID: 20% of pCd-2a, 6% of pCd-2b, 3% of pCd-2c; FlyWire: 32% of pCd-2a, 8% of pCd-2b, 13% of pCd-2c), the Murthy and Seung lab (FlyWire: 15% of pCd-2a, 11% of pCd-2b, 6% of pCd-2c), and Janelia tracers (FlyWire: 52% of pCd-2a, 14% of pCd-2b). We would like to thank the Jefferis labs (46%) and the Murthy and Seung labs (50%) for their contribution to tracing the t-shape Lgr3 + cells. The tracers who contributed include Nora Forknall, Lily Talley, Katie Stevens, Shanice Bailey, Tansy Yang, Istvan Taisz, Greg Jefferis, Lucas Encarnacion-Rivera, Austin T Burke, Yijie Yin, James Hebditch, Kyle Patrick Willie, Joshua Bañez, Philipp Schlegel, Dharini Sapkal, Irene Salgarella, Dhwani Patel, Ben Silverman, Bhargavi Parmar, Chitra Nair, Doug Bland, Laia Serratosa, Marta Costa, Nash Hadjerol, Regine Salem, Rey Adrian Candilada, Shirleyjoy Serona, Zairene Lenizo, Zeba Vohra, and Zepeng Yao.

## Additional information

### Funding

| Funder | Grant reference number | Author |
| --- | --- | --- |
| National Institutes of Health | R01DC013280 | Kristin Scott |

The funders had no role in study design, data collection and interpretation, or the decision to submit the work for publication.

### Author contributions

Meghan Laturney, Conceptualization, Data curation, Software, Formal analysis, Investigation, Visualization, Methodology, Writing - original draft, Writing - review and editing; Gabriella R Sterne, Data curation, Formal analysis, Visualization; Kristin Scott, Conceptualization, Supervision, Funding acquisition, Project administration, Writing - review and editing

### Author ORCIDs

Meghan Laturney  http://orcid.org/0000-0002-6428-5565
Gabriella R Sterne  http://orcid.org/0000-0002-7221-648X
Kristin Scott  http://orcid.org/0000-0003-3150-7210

Decision letter and Author response
Decision letter https://doi.org/10.7554/eLife.85117.sa1
Author response https://doi.org/10.7554/eLife.85117.sa2

## Additional files

### Supplementary files
• Supplementary file 1. Fly genotypes in figures. Table including the short and full genotypes of flies in the indicated figure.
• MDAR checklist

### Data availability
All data generated and analyzed during this study are included in the manuscript and supporting files.

The following previously published dataset was used:

| Author(s) | Year | Dataset title | Dataset URL | Database and Identifier |
|---|---|---|---|---|
| Zheng Z, Lauritzen JS, Perlman E, Robinson CG, Nichols M, Milkie D, Torrens O, Price J, Fisher CB, Sharifi N, Calle-Schuler SA, Kmecova L, Ali IJ, Karsh B, Trautman ET, Bogovic JA, Hanslovsky P, Jefferies GSXE, Kazhdan M, Khairy K, Saalfeld S, Fetter RD, Bock DD | 2018 | A Complete Electron Microscopy Volume of the Brain of Adult Drosophila melanogaster | https://catmaid-fafb.virtualflybrain.org/ | Adult Brain, catmaid |

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

## Appendix 1

**Appendix 1—key resources table**

| Reagent type (species) or resource | Designation | Source or reference | Identifiers | Additional information |
|---|---|---|---|---|
| Antibody | Anti-Brp (mouse monoclonal) | DSHB, University of Iowa, USA | DSHB Cat# nc82, RRID:AB_2314866 | 1/40 |
| Antibody | Anti-GFP (chicken polyclonal) | Thermo Fisher Scientific | Thermo Fisher Scientific Cat# A10262, RRID:AB_2534023 | 1/1000 |
| Antibody | Anti-chicken Alexa Fluor 488 (goat polyclonal) | Thermo Fisher Scientific | Thermo Fisher Scientific Cat# A11039, RRID:AB_2534096 | 1/500 |
| Antibody | Anti-mouse Alexa Fluor 647 (goat polyclonal) | Thermo Fisher Scientific | Thermo Fisher Scientific Cat# A21236, RRID:AB_2535805 | 1/500 |
| Antibody | Anti-HA (rabbit monoclonal) | Cell Signalling Technology | Cell Signaling Technology (C29F4) Cat #3724, RRID:AB_1549585 | 1/1000 |
| Antibody | Anti-rabbit Alexa Fluor 488 (goat polyclonal) | Thermo Fisher Scientific | Thermo Fisher Scientific Cat# A11034, RRID:AB_2576217 | 1/500 |
| Antibody | Anti-DYKDDDK epitope tag (rat monoclonal) | Novus Biologicals | Novus Biologicals, NBP1-06712, RRID:AB_1625981 | 1/1000 |
| Antibody | Anti-rat Alexa Fluor 568 (goat polyclonal) | Thermo Fisher Scientific | Thermo Fisher Scientific Cat# A11077, RRID#AB_2534121 | 1/500 |
| Antibody | Anti-dsRed (rabbit polyclonal) | Takara | Takara Bio Cat# 632496, RRID:AB_10013483 | 1/1000 |
| Antibody | Anti-rabbit Alexa Fluor 568 (goat polyclonal) | Thermo Fisher Scientific | Thermo Fisher Scientific Cat# A11011, RRID#AB_143157 | 1/500 |
| Chemical Compound, drug | All trans-Retinal | MilliporeSigma | Cat # R2500, CAS:116-31-4 | |
| Chemical Compound, drug | Sucrose | Thermo Fisher Scientific | Thermo Fisher Scientific:Cat# AAA1558336; CAS:57-50-1 | |
| Genetic reagent (*D. melanogaster*) | Wild-type, Canton-S | Bloomington Stock Center | | |
| Genetic reagent (*D. melanogaster*) | Gr64f-Gal4 | Bloomington Stock Center | RRID: BDSC_57669 | |
| Genetic reagent (*D. melanogaster*) | Gr64f-Gal4 | Bloomington Stock Center | RRID: BDSC_57668 | |

*Appendix 1 Continued on next page*

*Appendix 1 Continued*

| Reagent type (species) or resource | Designation | Source or reference | Identifiers | Additional information |
|---|---|---|---|---|
| Genetic reagent (*D. melanogaster*) | UAS-GCaMP6s(attP40) | Bloomington Stock Center | RRID: BDSC_42746 | |
| Genetic reagent (*D. melanogaster*) | UAS-GCaMP6s(VK00005) | Bloomington Stock Center | RRID: BDSC_42749 | |
| Genetic reagent (*D. melanogaster*) | UAS-CD8-tdTomato | *Thistle et al., 2012* | | |
| Genetic reagent (*D. melanogaster*) | ovoD1 | *Mével-Ninio et al., 1996* | | |
| Genetic reagent (*D. melanogaster*) | hs-bam | Bloomington Stock Center | RRID: BDSC_24636 | |
| Genetic reagent (*D. melanogaster*) | nsyb-Gal4 | Bloomington Stock Center | RRID: BDSC_51941 | |
| Genetic reagent (*D. melanogaster*) | UAS-dcr-2 | Bloomington Stock Center | RRID: BDSC_24650 | |
| Genetic reagent (*D. melanogaster*) | UAS-SPR-IR1 | *Yapici et al., 2008* | | |
| Genetic reagent (*D. melanogaster*) | ppk-gal4 | Bloomington Stock Center | RRID: BDSC_32079 | |
| Genetic reagent (*D. melanogaster*) | VT003280-gal4 (attP2) | *Feng et al., 2014* | | |
| Genetic reagent (*D. melanogaster*) | VT003280-LexA (attP2) | *Feng et al., 2014* | | |
| Genetic reagent (*D. melanogaster*) | Fru$^{P1-LexA}$ | *Mellert et al., 2010* | | |
| Genetic reagent (*D. melanogaster*) | 8xLexAop2-FLPL(attP40) | Bloomington Stock Center | RRID: BDSC_55820 | |
| Genetic reagent (*D. melanogaster*) | UAS >stop > KIR | Bloomington Stock Center | RRID: BDSC_67686 | |
| Genetic reagent (*D. melanogaster*) | UAS >stop > TNT | Bloomington Stock Center | RRID: BDSC_67690 | |
| Genetic reagent (*D. melanogaster*) | UAS >stop > CD8::GF | Bloomington Stock Center | RRID: BDSC_30125 | |
| Genetic reagent (*D. melanogaster*) | VT050405-p65ADZp(attP40) | *Tirian and Dickson, 2017* | | |

*Appendix 1 Continued on next page*

*Appendix 1 Continued*

| Reagent type (species) or resource | Designation | Source or reference | Identifiers | Additional information |
|---|---|---|---|---|
| Genetic reagent (*D. melanogaster*) | dsx-ZpGal4DBD | *Shirangi et al., 2016* | | |
| Genetic reagent (*D. melanogaster*) | pC1 split-GAL4, SS01491 | Bloomington Stock Center | RRID:BDSC_86830 | Full genotype: VT002064-p65ADZp(attP40); VT008469-ZpGal4DBD(attP2) |
| Genetic reagent (*D. melanogaster*) | VT002064-p65ADZp | Bloomington Stock Center | RRID:BDSC_72442 | |
| Genetic reagent (*D. melanogaster*) | vpo split-GAL4, SS50200 | Bloomington Stock Center | RRID:BDSC_86868 | Full genotype: 31D07-p65ADZp (attP40); 52F12-ZpGal4DBD (attP2) |
| Genetic reagent (*D. melanogaster*) | vpo split-GAL4, SS50795 | Bloomington Stock Center | RRID:BDSC_86869 | Full genotype: VT045670-p65ADZp (attP40); 52F12-ZpGal4DBD (attP2) |
| Genetic reagent (*D. melanogaster*) | oviIN split-GAL4, SS65422 | Bloomington Stock Center | RRID:BDSC_86837 | Full genotype: 68A10-p65ADZp (attP40); VT010054-ZpGal4DBD (attP2) |
| Genetic reagent (*D. melanogaster*) | oviIN split-GAL4, SS65423 | Bloomington Stock Center | RRID:BDSC_86838 | Full genotype: VT026347-p65ADZp (attP40); VT026035-ZpGal4DBD (attP2) |
| Genetic reagent (*D. melanogaster*) | oviDN split-GAL4, SS46540 | Bloomington Stock Center | RRID:BDSC_86832 | Full genotype: VT050660-p65ADZp (attP40); VT028160-ZpGal4DBD (attP2) |
| Genetic reagent (*D. melanogaster*) | oviDN split-GAL4, SS35666 | Bloomington Stock Center | RRID:BDSC_86831 | Full genotype: VT026873-p65ADZp (attP40); VT040574-ZpGal4DBD (attP2) |
| Genetic reagent (*D. melanogaster*) | 20xUAS-IVS-GtACR1-EYFP9attP2 | *Mohammad et al., 2017* | | |
| Genetic reagent (*D. melanogaster*) | 20xUAS-CsChrimson::mVenus (attP18) | Bloomington Stock Center | RRID: BDSC_55134 | |
| Genetic reagent (*D. melanogaster*) | MCFO | Bloomington Stock Center, *Nern et al., 2015* | RRID:BDSC_64088 | |
| Genetic reagent (*D. melanogaster*) | VT004974-p65ADZp (attP40) | Bloomington Stock Center | RRID: BDSC_71475 | |
| Genetic reagent (*D. melanogaster*) | R69H11-ZpGal4DBD (attP2) | Bloomington Stock Center | RRID: BDSC_69819 | |
| Genetic reagent (*D. melanogaster*) | VT023823-p65ADZp (attP40) | *Tirian and Dickson, 2017* | | |

*Appendix 1 Continued on next page*

*Appendix 1 Continued*

| Reagent type (species) or resource | Designation | Source or reference | Identifiers | Additional information |
|---|---|---|---|---|
| Genetic reagent (*D. melanogaster*) | VT027804-p65ADZp (attP40) | Bloomington Stock Center | RRID: BDSC_72041 | |
| Genetic reagent (*D. melanogaster*) | VT057275-p65ADZp (attP40) | Bloomington Stock Center | RRID: BDSC_71664 | |
| Genetic reagent (*D. melanogaster*) | R9B05-p65ADZp (attP40) | Bloomington Stock Center | RRID: BDSC_70519 | |
| Genetic reagent (*D. melanogaster*) | R22D06-ZpGal4DBD (attP2) | *Dionne et al., 2018* | | |
| Genetic reagent (*D. melanogaster*) | R29G03-p65ADZp (attP40) | Bloomington Stock Center | RRID: BDSC_75746 | |
| Genetic reagent (*D. melanogaster*) | R60C09-p65ADZp (attP40) | Bloomington Stock Center | RRID: BDSC_69750 | |
| Genetic reagent (*D. melanogaster*) | R19B09-gal4 (attP2) | Bloomington Stock Center | RRID: BDSC_48840 | |
| Genetic reagent (*D. melanogaster*) | R19B09-LexA (attP40) | Bloomington Stock Center | RRID: BDSC_52539 | |
| Genetic reagent (*D. melanogaster*) | UAS >stop > csChrimson w-,UAS >stop > csChrimson (attP18) | *Wu et al., 2016* | | |
| Genetic reagent (*D. melanogaster*) | UAS-Chrimson88 | *Klapoetke et al., 2014* | | |
| Genetic reagent (*D. melanogaster*) | LexAop-GCaMP6s | *Broussard et al., 2018* | | |
| Genetic reagent (*D. melanogaster*) | R40F04-LexA (attP40) | Bloomington Stock Center | RRID: BDSC_54785 | |
| Software, Algorithm | Fiji | https://fiji.sc/ | RRID: SCR_002285 | |
| Software, Algorithm | GraphPad Prism | Graphpad Software; https://www.graphpad.com/scientific-software/prism/ | RRID:SCR_002798 | |
| Software, Algorithm | Python | Python Software Foundation; https://www.python.org/downloads/ | RRID:SCR_008394 | |
| Software, Algorithm | Flywire | Flywire; https://flywire.ai/ | RRID:SCR_019205 | |

*Appendix 1 Continued on next page*

*Appendix 1 Continued*

| Reagent type (species) or resource | Designation | Source or reference | Identifiers | Additional information |
|---|---|---|---|---|
| Software, Algorithm | Adobe Illustrator | Adobe Software; https://www.adobe.com/products/illustrator.html | RRID:SCR_010279 | |
| Software, Algorithm | CATMAID | *Saalfeld et al., 2009*; https://catmaid.org | RRID:SCR_006278 | |
| Software, Algorithm | CAVE (connectome annotation versioning engine) | *Buhmann et al., 2021*, *Heinrich et al., 2018* | | |
| Software, Algorithm | R Project for Statistical Computing | *Dessau and Pipper, 2008*; www.r-project.org | RRID:SCR_001905 | |
| Software, Algorithm | SciPy package | *Virtanen et al., 2020*; https://scipy.org/ | RRID:SCR_008058 | |
| Software, Algorithm | CircuitCatcher | *Bushey, 2019* | | |
| Other | FLIC | *Ro et al., 2014*; https://www.sablesys.com/products/classic-line/flic_drosophila_behavior_system/ | | |
| Peptide, recombinant protein | Alexa Fluor 633 Phalloidin | Thermo Fisher Scientific | Thermo Fisher Scientific Cat# A22284 | 1/500 |

