## [Editor Report]

After mating, animals show a repertoire of behavioural changes. In flies, this includes an increase in egg-laying, salt, and food (particularly protein) consumption, and a concomitant decrease in sexual receptivity. This valuable study compellingly shows that flies also have an increased sugar appetite and they identify the central brain circuitry that controls this increase in the mated condition.

---

## [Decision Letter]

**Decision letter after peer review:**

Thank you for submitting your article "Mating activates neuroendocrine pathways signaling hunger in *Drosophila* females" for consideration by *eLife*. Your article has been reviewed by 3 peer reviewers, including Sonia Sen as the Reviewing Editor and Reviewer #1, and the evaluation has been overseen by K VijayRaghavan as the Senior Editor. The following individual involved in the review of your submission has agreed to reveal their identity: Kenta Asahina (Reviewer #2).

Essential revisions:

We appreciate the overall rigour and importance of this manuscript. We have a few suggestions for the authors that we think will strengthen their claims.

1. The writing:

a. We note that the manuscript only briefly refers to the extensive literature on protein feeding and its modulation in response to mating. We suggest that the authors explicitly refer to this body of literature in the introduction and discuss their findings in its context.

b. We also suggest that the authors tone down the claim that the Lgr3+ neurons are a hub that integrates mating and hunger status to modulate feeding since this integration per se hasn't been tested.

2. The role of pCd2 and/or Lgr3+ in sugar and protein feeding. Our suggestion to the authors is to recapitulate the virgin circuit state in the mated females by activating pCd2 and/or silencing Lgr3+. Perhaps the authors have already done these manipulations, which should result in the suppression of sugar feeding – do they?

3. The role of mating: One of the main claims of the manuscript is that mating increases sugar appetite. We felt that the circuit's requirement in a mating-dependent manner has not been convincingly demonstrated. The authors have performed manipulations that, in the virgin, drive the circuit to a mated state (silenced pCd2 and activated Lgr3+ ) and demonstrated an increase in sugar consumption. We felt that interpretations from these manipulations would be more valuable in the context of similar manipulations in the mated state. If their model is correct, we anticipate that these manipulations should not result in changes in sugar feeding. (If they do, it might just suggest that optogenetics does not recapitulate SP-mediated activation of the circuit.)

3. Verification of the R40F04-LexA line. Since this is a new and potentially valuable reagent, it will be nice to describe its expression pattern thoroughly.

In addition to this, there are several more detailed comments in the reviews below and we urge the authors to address them wherever possible.

*Reviewer #1 (Recommendations for the authors):*

We very much enjoyed reading the manuscript for its clarity of data and its presentation.

1. Given the vast amount of literature on the increase in protein preference post-mating, it is important to discuss these data in the context of those findings. In those data, flies are presented with a choice of sugars and proteins and seem to prefer proteins. But the data in this manuscript, and Carvalho et al. 2006, suggest that this might be a general increase in hunger overall. I would urge the authors to discuss this explicitly.

2. In the circuit dissection experiments, one would expect that activating pCds or inactivating the Lgr3+ neurons in the mated condition should attenuate the sugar appetite. Does it?

3. I liked the authors' interpretation of their circuit mapping – that the mated state is essentially a heightened hunger state. If this is true, these manipulations should also influence protein hunger. Can the authors show this?

*Reviewer #2 (Recommendations for the authors):*

I have 3 recommendations on the current manuscript, 2 on an experiment, and 1 on the discussion.

Regarding the experiment:

1) any evidence that functions of pCd-2 and Lgr3 neurons are under influence of mating status will greatly strengthen the authors' conclusion that their role is to promote sucrose consumption after mating. Since direct measurement of baseline neuronal activities in pCd-2 and Lgr3 neurons is likely difficult in the *Drosophila* brain, one possible experiment is to test both the function (Figure 5I, J; Figure 6G) and/or the strength of the neural connection (Figure 5G, H; Figure 6D, E) is mating status-dependent. Currently, these experiments were performed only in virgin females. If the authors' model is correct (which I think is very likely), activation of pCd-2 (or their upstreams) should suppress sucrose consumption in mated females, but (at least) less so in virgin females (in which pCd-2 neurons are expected to be already active). In contrast, the inhibition of pCd-2 neurons may not change the sucrose consumption in mated females (in which pCd-2 neurons are already silenced). Moreover, stronger stimulation of pC1 and pC2-d may be required to excite pCd-2 and Lgr3, respectively, in mated females. Such mating status-dependent change in neural function can provide strong support for their hypothesis. No difference between mated and virgin females, in contrast, suggests that the levels of post-mating inhibition are likely much weaker than the optogenetic manipulation, suggesting nuanced modulation by the Sex Peptide. Either result would be valuable for follow-up studies.

2) Provide evidence that R40F04-LexA indeed labels pC1a neurons, ideally by the overlap of expression with known pC1a split GAL4 lines. While the authors' argument that pC1a and pC2-a/b (labeled by the split GAL4s) are synaptically connected is convincing, the functional imaging experiment in Figure 5G, H is confounded by potential side effect by the activation of non-pC1 neurons (which are many, according to the image sample on FlyLight database). Since R40F04-LexA has not been previously used as a proxy of pC1a, a basic characterization of its expression pattern will help readers better interpret the data (along with its caveat).

In the discussion, I would like to suggest that authors better incorporate previous publications regarding the mating status-dependent changes in female feeding behaviors. For instance, Walker et al., Curr. Biol. (2015) showed that the SPSN-SAG axis controls both yeast and salt preference among mated females. This result is consistent with data in Steck et al., *eLife* (2018), which showed that the sensitivity of yeast-sensing GRNs remains unchanged after mating (similar to the sugar-sensing GRNs in this study). Discussions on whether pCd-2 also controls increased intake of these compounds can inspire readers in the field. Also, Hussain et al., PLoS Biol. (2016) showed that GRNs' sensitivity to polyamine is increased after mating, in contrast to the relatively stable sensitivity of sensory neurons shown in both this manuscript and Steck et al. Mating may exert its effect on different behaviors or sensory modalities in a distinct manner, which is worth mentioning given the Sex Peptide has received intense (and potentially outsized) attention in the context of mating-dependent behavioral changes. Without due discussion of these previous studies, the authors' assertion that the current study "examine(d) neural mechanisms for appetitive changed in mated females" (line 52) may appear premature.

*Reviewer #3 (Recommendations for the authors):*

Overall, the major claims of this paper are interesting, important, and well-supported by the data. I would just note a few caveats:

1. The authors do not perform silencing of Lgr3+ mBNs to show that they're required for increased feeding in mated females. This would be important to demonstrate whether these cells really have a key role in this post-mating change. Similarly, the role of pCd-2 neurons was shown through silencing in virgins to mimic mating; their role could be strengthened by activating them in mated females and showing a decrease in feeding.

2. It has not been shown that the function of Lgr3/Dilp8 in regulating feeding occurs in the same median bundle cells that are affected by mating. This could be tested by knocking down Lgr3 specifically in these cells. Without this data, the major claims of the paper are still supported but the notion that the same cells integrate hunger and mating signals is somewhat speculative.

3. Since a major implication of this paper is that mating and hunger signals converge on the same cells and may thus interact, it is important for the authors to be clear about whether virgin and mated females are being compared in the fed or starved state. Does mating affect feeding in just one of these states, or in both states? The authors may have the data to answer this question (e.g. I think the FLIC assays were performed in starved flies and the CAFE assays were performed in fed flies), but it is not discussed in the paper.

4. A critical piece of this story is to know that the "Lgr3+" mBNs that are being manipulated actually express Lgr3. This is not obvious or explicitly discussed, but it seems that co-expression of the driver lines used here with an Lgr3 line may have been performed in a previous study (Meissner et al., 2016). The authors should clarify this and be explicit about what proportion of Lgr3+ mBNs they think they are targeting, and also what proportion of neurons they are targeting likely express Lgr3.

5. The picture of the "off-target" SAGb neuron in Figure 3 – —figure supplement 3A doesn't really look like its pictures in panel B of the same figure, where it is labeled by different driver lines; in fact the additional lines look more like SAG than SAGb. I am not sure how the authors know that these other driver lines are labeling the same neuron as the "off-target" neuron in the SAG-SS3 line.

6. The authors describe Lgr3 as a hunger signal and imply that it is independent of mating, but, interestingly, Lgr3 is only expressed in females. This implies that it is not a generic hunger signal but has some specialized role. (One of the cited papers, Yeom et al., mentions that Dilp8 is highly expressed in the female ovary.) The authors may want to speculate about why Lgr3 is sex-specific and not simply frame Lgr3 as a generic hunger signal.

In addition to the above comments there are some additional comments that should be addressed but, for the most part, are not central to the major claims of the paper.

1. Several comments related to the feeding assays:

a) It seems like the flies in your CAFE assay are consuming much less than in previous studies, which reported that female flies consume something on the order of 1 µl/day (e.g. Ja et al., 2007; Farhadian et al., 2012). Is there any explanation for why your flies are consuming much less? I know you're using 50 mM sucrose which is less than most other studies, but your FLIC data (Figure 1D) suggests that total consumption isn't highly dependent on concentration.

b) It seems strange that cumulative consumption over 6 hours is a straight line (Figure 1B). Given that these flies were starved, I would expect to see high consumption in the first hour that then decreases substantially. Also, feeding bouts are typically clustered in the morning and evening, and not distributed equally throughout the day. Any thoughts on this?

c) It seems strange that changing the sugar concentration more than 10-fold results in hardly any difference in feeding (Figure 1D-F). I might expect no difference if measuring over an entire day when homeostatic processes dominate over taste, but this is a 20 min assay. Also, the authors use the lack of difference in bout number between mated and virgin females as a justification for why feeding initiation (PER) is unlikely to be different. However, the bout number is also similar across sugar concentrations whereas PER is very different (at least for fed flies), which casts doubt on that logic.

2. The text states: "Thus, like hungry flies (Itskov et al., 2014), mated females consume more by engaging in longer feeding times rather than by initiating more feeding events." This seems misleading since the paper by Itskov et al. shows a huge difference in the number of sips for fed and starved flies, whereas the sip duration did not differ (Figure 5C and 6E).

3. What was the rationale for focusing on pCd-2a and pCd-2b rather than pCd-2c?

4. In Figure 5 – Figure supplement 1, the driver lines seem to label a few cells other than pCd-2 (e.g. in the optic lobe, abdominal ganglion, or along the surface of the brain), so it seems strange that in most cases the male brains completely lack any expression, even of the non-pCd-2 neurons.

5. As noted in Point # 2 of the Public Review, I think that the data show a possible convergence of mating and hunger signals in the same cells, but it hasn't been shown definitively. Given this, I think the title is a bit speculative. I also think this sentence in the abstract is not really supported by the data: "Together, this circuit transforms the mated signal into a long-term hunger signal."

6. A functional connection between SAG and pCd-2 neurons was not tested here. This is not critical, but it would be interesting to not only confirm this connection but to test how the co-activation of pC1 and SAG affects the pCd-2 neurons. This might hint at why the circuit evolved to have both pC1 and SAG connected to pCd-2 rather than just one or the other. If the authors have ideas on this they could speculate in the Discussion.

---

## [Author Response]

Essential revisions:We appreciate the overall rigour and importance of this manuscript. We have a few suggestions for the authors that we think will strengthen their claims.1. The writing:a. We note that the manuscript only briefly refers to the extensive literature on protein feeding and its modulation in response to mating. We suggest that the authors explicitly refer to this body of literature in the introduction and discuss their findings in its context.

We agree with the reviewers that our findings should be discussed in more detail within the context of what is already established in the field. Text was changed in the introduction (Line 32-37, 56-71) to expand on the mated related changes in protein and salt consumption.

Text was also added (3 paragraphs) to the discussion to discuss our findings within a larger context of feeding regulation (Line 286-321).

b. We also suggest that the authors tone down the claim that the Lgr3+ neurons are a hub that integrates mating and hunger status to modulate feeding since this integration per se hasn't been tested.

To tone down the language claiming that Lgr3+ neurons integrate these two signals, the following sentence was removed from the discussion: “The sexually dimorphic Lgr3+ neurons are the nexus integrating mating status and neuroendocrine hunger signaling.” The following was added:

“Importantly, it has not been established how Dilp8 influences the activity of Lgr3+ MBNs. Regardless, together these data suggest a proposed model for how Lgr3+ mBN neurons integrate signals from the mating status and hunger/satiety systems (Figure 7). (Line 278-281)

2. The role of pCd2 and/or Lgr3+ in sugar and protein feeding. Our suggestion to the authors is to recapitulate the virgin circuit state in the mated females by activating pCd2 and/or silencing Lgr3+. Perhaps the authors have already done these manipulations, which should result in the suppression of sugar feeding – do they?

The experiments presented in the manuscript show evidence that manipulating the activity of the mating status circuit (SPSNs-SAG-pC1), a mating status output neuron cell type (pCd2), and Lgr3+ MBNs in virgin females to mimic the mated state was sufficient to recapitulate the postmated behavior of increased sucrose consumption. However, an additional question of necessity was not originally investigated: are pCd-2 or Lgr3+ neurons necessary for this postmating response? We addressed this in two ways.

First, we activated pCd-2a and pCd-2b in mated females, mimicking the virgin state, and found that activating pCd-2a (but not pCd-2b) significantly reduces sugar intake recapitulating the virgin phenotype in a mated female (Figure 5 figure supplement 4). This line was added to the manuscript (Line 209-217):

“Next, we activated pCd-2a or pCd-2b in a mated female to mimic the virgin state to test if this cell type is necessary for the postmated increase in sugar intake. We found that mated females with pCd-2a artificially activated significantly reduced sugar intake compared to controls, recapitulating the virgin phenotype in mated females (Figure 5 —figure supplement 4). However, mated females with pCd-2b artificially activated did not differ from controls (Figure 5 —figure supplement 4), which may be a reflection of the few number of synapses from SAG and pC1 (Figure 5A).”

Second, we set out to silence Lgr3+ MBNs in mated females, mimicking the virgin state. We were not able to acutely silence via optogenetics for 2 reasons. First, we are not aware that the line UAS>stop>GtACR1 exists. Second, we were not able to use R19B09-gal4 (the line we used to access Lgr3 MBNs) with UAS-GtACR1 as this GAL4 line also labels neurons in the SEZ (Liao and Nässel, Front Endocrinol, 2020), which would make interpretation of these results impossible. We then used a constitutive silencer with an intersectional method (UAS>stop>KIR with R19B09-gal4 and Fru-gal4). We found that virgin and mated females with Lgr3+ MBNs silenced did not differ in sucrose consumption suggesting that the mating status circuit activates these neurons after mating to induce sucrose feeding. However, one of the control lines also failed to show this increase making these results difficult to interpret. Although we have included this as a Author response image 1, we are not including this dataset in the manuscript.

**Author response image 1. sa2fig1:** Total drinking time in FLIC in a 20 minute assay of females of indicated genotype and mating state. White circles represent virgin and black circles indicate mated, n = 14-17. Data was analyzed with a Two-Way ANOVA and revealed a significant interaction of genotype and mating status, F(2,88)=9.271, p<0.001. Bonferroni posthoc tests indicated a significant increase in feeding after mating in only one control genotype (uas>stop>KIR, Fru-LexA), *p<0.05, ***p<0.001, ns = not significant. Scatterplot show mean +/- s.e.m.

3. The role of mating: One of the main claims of the manuscript is that mating increases sugar appetite. We felt that the circuit's requirement in a mating-dependent manner has not been convincingly demonstrated. The authors have performed manipulations that, in the virgin, drive the circuit to a mated state (silenced pCd2 and activated Lgr3+ ) and demonstrated an increase in sugar consumption. We felt that interpretations from these manipulations would be more valuable in the context of similar manipulations in the mated state. If their model is correct, we anticipate that these manipulations should not result in changes in sugar feeding. (If they do, it might just suggest that optogenetics does not recapitulate SP-mediated activation of the circuit.)

Using functional imaging we established that pCd-2 neurons are activated by pC1, suggesting that they are likely activate in a virgin female and silenced in the mated state. Moreover, we show that Lgr3+ MBNs are silenced by pCd-2 neurons and therefore it is very likely that they are silent (or at least less active) in a virgin female and active in the mated state. We then mimicked the mated state in both cell types in virgin

females to induce the mated phenotype and observed increases in sucrose consumption. We believe that this evidence demonstrates the circuits involvement in mating-dependent modulation of sugar feeding.

To explore if artificially silencing pCd2 with GtAC1 and exposure to constant green light (to induce a mated state) imitated the mated state firing pattern, we silenced these neurons in an already mated female (using the same methods) and observed sucrose intake. We included these results in the manuscript (Line 203-209).

“To explore if optogenetic manipulation of pCd-2a or pCd-2b mimicked the mated state, we silenced these neurons in already mated females. No difference in sucrose consumption would suggest that artificial silencing imitates the mated firing pattern. Instead, we found that mated females with pCd-2a or pCd-2b silenced further increased sucrose consumption (Figure 5—figure supplement 4), suggesting that optogenetic inhibition is stronger than mating-induced inhibition. This is reasonable since the mated state likely decreases activity of the mating status circuit rather than silences it completely (Feng et al. 2014).”

4. Verification of the R40F04-LexA line. Since this is a new and potentially valuable reagent, it will be nice to describe its expression pattern thoroughly.

We imaged the expression pattern of R40F04-LexA and pC1-SS1 to provide evidence that the R40F04-LexA does indeed label this cell type. We included these results in the manuscript:

“To test if pCd-2a and pCd-2b are functionally connected with pC1, we optogenetically activated pC1 using R40F04-LexA (Figure 5 —figure supplement 2) while

simultaneously measuring calcium responses in either pCd-2a or pCd-2b neurons. “ (Line 192-195)

“We verified that R40F04-LexA labels pC1 and describe the expression pattern (see Figure 5- supplemental figure 2).” (Line 452-453)

Reviewer #1 (Recommendations for the authors):We very much enjoyed reading the manuscript for its clarity of data and its presentation.1. Given the vast amount of literature on the increase in protein preference post-mating, it is important to discuss these data in the context of those findings. In those data, flies are presented with a choice of sugars and proteins and seem to prefer proteins. But the data in this manuscript, and Carvalho et al. 2006, suggest that this might be a general increase in hunger overall. I would urge the authors to discuss this explicitly.

As addressed in Essential Revisions #1, we agree with the reviewers that our findings should be discussed in more detail within the context of what is already established in the field. Text was changed in the introduction (Line 32-37, 56-71) to expand on the mated related changes in protein and salt consumption.

Text was also added (3 paragraphs) to the discussion to discuss our findings within a larger context of feeding regulation (Line 286-321).

2. In the circuit dissection experiments, one would expect that activating pCds or inactivating the Lgr3+ neurons in the mated condition should attenuate the sugar appetite. Does it?

As addressed in Essential Revisions #2, we activated pCd-2a and pCd-2b in mated females, mimicking the virgin state, and found that activating pCd-2a (but not pCd-2b) significantly reduces sugar intake recapitulating the virgin phenotype in a mated female (Figure 5 figure supplement 4). This line was added to the manuscript (Line 209-217):

“Next, we activated pCd-2a or pCd-2b in a mated female to mimic the virgin state to test if this cell type is necessary for the postmated increase in sugar intake. We found that mated females with pCd-2a artificially activated significantly reduced sugar intake compared to controls, recapitulating the virgin phenotype in mated females (Figure 5 —figure supplement 4). However, mated females with pCd-2b artificially activated did not differ from controls (Figure 5 —figure supplement 4), which may be a reflection of the few number of synapses from SAG and pC1 (Figure 5A).”

3. I liked the authors' interpretation of their circuit mapping – that the mated state is essentially a heightened hunger state. If this is true, these manipulations should also influence protein hunger. Can the authors show this?

Based on the experiments that we have performed, we cannot determine if activating Lgr3+ fru+ MBNs would increase all food intake or is specific to sucrose. This is an interesting question that would require us to develop additional assays and perform additional experiments that are beyond the scope of this study. We include a paragraph in the discussion on this topic: “Although mating status impinges on circuits that either process nutrient sensing signals or regulate feeding circuits, there is no shared circuit yet identified for elevated nutrient intake of sugar and yeast in mated females. Beyond the three-tier mating status circuit, no overlap between areas of the brain modulating postmated increases in protein (Liu et al. 2017, Münch et al. 2022) and sugar consumption (this manuscript) have been found. It remains to be elucidated whether the circuit described here specifically modulates post-mating changes in sucrose consumption or whether it impacts the intake of other nutrients, including protein and salt.” (Line 315-321).

Reviewer #2 (Recommendations for the authors):I have 3 recommendations on the current manuscript, 2 on an experiment, and 1 on the discussion.Regarding the experiment:1) any evidence that functions of pCd-2 and Lgr3 neurons are under influence of mating status will greatly strengthen the authors' conclusion that their role is to promote sucrose consumption after mating. Since direct measurement of baseline neuronal activities in pCd-2 and Lgr3 neurons is likely difficult in the *Drosophila* brain, one possible experiment is to test both the function (Figure 5I, J; Figure 6G) and/or the strength of the neural connection (Figure 5G, H; Figure 6D, E) is mating status-dependent. Currently, these experiments were performed only in virgin females. If the authors' model is correct (which I think is very likely), activation of pCd-2 (or their upstreams) should suppress sucrose consumption in mated females, but (at least) less so in virgin females (in which pCd-2 neurons are expected to be already active). In contrast, the inhibition of pCd-2 neurons may not change the sucrose consumption in mated females (in which pCd-2 neurons are already silenced). Moreover, stronger stimulation of pC1 and pC2-d may be required to excite pCd-2 and Lgr3, respectively, in mated females. Such mating status-dependent change in neural function can provide strong support for their hypothesis. No difference between mated and virgin females, in contrast, suggests that the levels of post-mating inhibition are likely much weaker than the optogenetic manipulation, suggesting nuanced modulation by the Sex Peptide. Either result would be valuable for follow-up studies.

The reviewer brings up two points addressed in the Essential Revisions (#2 and #3).

Wwe activated pCd-2a and pCd-2b in mated females, mimicking the virgin state, and found that activating pCd-2a (but not pCd-2b) significantly reduces sugar intake recapitulating the virgin phenotype in a mated female (Figure 5 figure supplement 4). This line was added to the manuscript (Line 209-217):

“Next, we activated pCd-2a or pCd-2b in a mated female to mimic the virgin state to test if this cell type is necessary for the postmated increase in sugar intake. We found that mated females with pCd-2a artificially activated significantly reduced sugar intake compared to controls, recapitulating the virgin phenotype in mated females (Figure 5 —figure supplement 4). However, mated females with pCd-2b artificially activated did not differ from controls (Figure 5 —figure supplement 4), which may be a reflection of the few number of synapses from SAG and pC1 (Figure 5A).”

To explore if artificially silencing pCd2 with GtAC1 and exposure to constant green light (to induce a mated state) imitated the mated state firing pattern, we silenced these neurons in an already mated female (using the same methods) and observed sucrose intake. We included these results in the manuscript (Line 203-209).

“To explore if optogenetic manipulation of pCd-2a or pCd-2b mimicked the mated state, we silenced these neurons in already mated females. No difference in sucrose consumption would suggest that artificial silencing imitates the mated firing pattern. Instead, we found that mated females with pCd-2a or pCd-2b silenced further increased sucrose consumption (Figure 5—figure supplement 4), suggesting that optogenetic inhibition is stronger than mating-induced inhibition. This is reasonable since the mated state likely decreases activity of the mating status circuit rather than silences it completely (Feng et al. 2014).”

2) Provide evidence that R40F04-LexA indeed labels pC1a neurons, ideally by the overlap of expression with known pC1a split GAL4 lines. While the authors' argument that pC1a and pC2-a/b (labeled by the split GAL4s) are synaptically connected is convincing, the functional imaging experiment in Figure 5G, H is confounded by potential side effect by the activation of non-pC1 neurons (which are many, according to the image sample on FlyLight database). Since R40F04-LexA has not been previously used as a proxy of pC1a, a basic characterization of its expression pattern will help readers better interpret the data (along with its caveat).

As addressed in the Essential Revisions #4, we imaged the expression pattern of R40F04-LexA and pC1-SS1 to provide evidence that the R40F04-LexA does indeed label this cell type. We included these results in the manuscript:

“To test if pCd-2a and pCd-2b are functionally connected with pC1, we optogenetically activated pC1 using R40F04-LexA (Figure 5 —figure supplement 2) while

simultaneously measuring calcium responses in either pCd-2a or pCd-2b neurons. “ (Line 192-195)

“We verified that R40F04-LexA labels pC1 and describe the expression pattern (see Figure 5- supplemental figure 2).” (Line 452-453)

In the discussion, I would like to suggest that authors better incorporate previous publications regarding the mating status-dependent changes in female feeding behaviors. For instance, Walker et al., Curr. Biol. (2015) showed that the SPSN-SAG axis controls both yeast and salt preference among mated females. This result is consistent with data in Steck et al., eLife (2018), which showed that the sensitivity of yeast-sensing GRNs remains unchanged after mating (similar to the sugar-sensing GRNs in this study). Discussions on whether pCd-2 also controls increased intake of these compounds can inspire readers in the field. Also, Hussain et al., PLoS Biol. (2016) showed that GRNs' sensitivity to polyamine is increased after mating, in contrast to the relatively stable sensitivity of sensory neurons shown in both this manuscript and Steck et al. Mating may exert its effect on different behaviors or sensory modalities in a distinct manner, which is worth mentioning given the Sex Peptide has received intense (and potentially outsized) attention in the context of mating-dependent behavioral changes. Without due discussion of these previous studies, the authors' assertion that the current study "examine(d) neural mechanisms for appetitive changed in mated females" (line 52) may appear premature.

As addressed in Essential Revisions #1, we agree with the reviewers that our findings should be discussed in more detail within the context of what is already established in the field. Text was changed in the introduction (LINE 32-37, 56-71) to expand on the mated related changes in protein and salt consumption.

Text was also added (3 paragraphs) to the discussion to discuss our findings within a larger context of feeding regulation (LINE 286-321).

Reviewer #3 (Recommendations for the authors):Overall, the major claims of this paper are interesting, important, and well-supported by the data. I would just note a few caveats:1. The authors do not perform silencing of Lgr3+ mBNs to show that they're required for increased feeding in mated females. This would be important to demonstrate whether these cells really have a key role in this post-mating change. Similarly, the role of pCd-2 neurons was shown through silencing in virgins to mimic mating; their role could be strengthened by activating them in mated females and showing a decrease in feeding.

As addressed in Essential Revisions #2, we activated pCd-2a and pCd-2b in mated females, mimicking the virgin state, and found that activating pCd-2a (but not pCd-2b) significantly reduces sugar intake recapitulating the virgin phenotype in a mated female (Figure 5 figure supplement 4). This line was added to the manuscript (Line 209-217): “Next, we activated pCd-2a or pCd-2b in a mated female to mimic the virgin state to test if this cell type is necessary for the postmated increase in sugar intake. We found that mated females with pCd-2a artificially activated significantly reduced sugar intake compared to controls, recapitulating the virgin phenotype in mated females (Figure 5 —figure supplement 4). However, mated females with pCd-2b artificially activated did not differ from controls (Figure 5 —figure supplement 4), which may be a reflection of the few number of synapses from SAG and pC1 (Figure 5A).”

2. It has not been shown that the function of Lgr3/Dilp8 in regulating feeding occurs in the same median bundle cells that are affected by mating. This could be tested by knocking down Lgr3 specifically in these cells. Without this data, the major claims of the paper are still supported but the notion that the same cells integrate hunger and mating signals is somewhat speculative.

We agree with the reviewer and appreciate their critical feedback. The work that showed Lgr3 and dilp8 act as satiety factors was accomplished with mutants and therefore the model we propose is speculative. The experiment that the reviewer suggested cannot be performed as the reagents do not exist. We used an intersectional method to access the Lgr3+ median bundle cells: R19B09-gal4 in combination with FruLexA, LexAop-FLPL. In order to knockdown Lgr3 specifically in median bundle cells, we would need a tool that conditionally expressed RNAi in the presence of a flipase (UAS>stop>Lgr3-RNAi). This conditional UAS line does not exist.

Instead, as addressed in Essential Revisions #1b, we tone down the language claiming that Lgr3+ neurons integrate these two signals, the following sentence was removed from the discussion: “The sexually dimorphic Lgr3+ neurons are the nexus integrating mating status and neuroendocrine hunger signaling.” The following was added:

“Importantly, it has not been established how Dilp8 influences the activity of Lgr3+ MBNs. Regardless, together these data suggest a proposed model for how Lgr3+ mBN neurons integrate signals from the mating status and hunger/satiety systems (Figure 7). (LINE 278-281)

3. Since a major implication of this paper is that mating and hunger signals converge on the same cells and may thus interact, it is important for the authors to be clear about whether virgin and mated females are being compared in the fed or starved state. Does mating affect feeding in just one of these states, or in both states? The authors may have the data to answer this question (e.g. I think the FLIC assays were performed in starved flies and the CAFE assays were performed in fed flies), but it is not discussed in the paper.

FLIC assays were performed on starved flies as they are shorter in length (20 minutes or 4 hours) compared to CAFE assays which are 24 hours. Starvation is done with shorter assays to encourage the flies to consume and is not required in longer assays as they have more time to exhibit feeding behavior. The behavioral assays are described in the Methods (Line 343-375).

4. A critical piece of this story is to know that the "Lgr3+" mBNs that are being manipulated actually express Lgr3. This is not obvious or explicitly discussed, but it seems that co-expression of the driver lines used here with an Lgr3 line may have been performed in a previous study (Meissner et al., 2016). The authors should clarify this and be explicit about what proportion of Lgr3+ mBNs they think they are targeting, and also what proportion of neurons they are targeting likely express Lgr3.

We agree that transparency regarding the rationale for using R19B09 driver lines to label the Lgr3+ neurons is important. We added a note to see methods in the Figure 6 caption (LINE 1053) and a thorough description in the methods (LINE 536-559).

5. The picture of the "off-target" SAGb neuron in Figure 3 – —figure supplement 3A doesn't really look like its pictures in panel B of the same figure, where it is labeled by different driver lines; in fact the additional lines look more like SAG than SAGb. I am not sure how the authors know that these other driver lines are labeling the same neuron as the "off-target" neuron in the SAG-SS3 line.

Our rationale of how we generated the SAGb split-GAL4 lines can be found in the methods (LINE 570-581). We added the following statement: “Expression pattern of the split GAL4 was then compared to SAGb. We confirmed the cells match in morphology with the following characteristics: the location of the soma, ladder rung arborizations in the ventral nerve cord, lack of mid- ventral nerve cord branches, bowing ascending neurites in the SEZ, and arborizations in the pars intercerebralis.” (LINE 577-581). And we modified Figure 3 —figure supplement 3 to showcase these morphological characteristics.

6. The authors describe Lgr3 as a hunger signal and imply that it is independent of mating, but, interestingly, Lgr3 is only expressed in females. This implies that it is not a generic hunger signal but has some specialized role. (One of the cited papers, Yeom et al., mentions that Dilp8 is highly expressed in the female ovary.) The authors may want to speculate about why Lgr3 is sex-specific and not simply frame Lgr3 as a generic hunger signal.

We modified the text to better represent that Lgr3 is not female specific but sexually dimorphic and expressed in fruitless expressing neurons, poised be involved in female specific behavior. The text now reads: “Lgr3 expression is sexually dimorphic, with a greater number of cells in the central brain and abdominal ganglia of females (Meissner et al. 2016), consistent with a role in female behavior” (LINE 227-229).

Lgr3 is not exclusively expressed females. Instead, there are more Lgr3+ neurons in females than in males (Meissner et al. 2016). Moreover, Dilp8 and Lgr3 influence feeding regulation in males (Yeom et al. 2021). Together, this indicates that there is not likely a sexually dimorphic function of this system. However, it is likely that the additional Lgr3+ fru+ MBNs in the female function to modulate feeding behavior via mating status.

In addition to the above comments there are some additional comments that should be addressed but, for the most part, are not central to the major claims of the paper.1. Several comments related to the feeding assays:a) It seems like the flies in your CAFE assay are consuming much less than in previous studies, which reported that female flies consume something on the order of 1 µl/day (e.g. Ja et al., 2007; Farhadian et al., 2012). Is there any explanation for why your flies are consuming much less? I know you're using 50 mM sucrose which is less than most other studies, but your FLIC data (Figure 1D) suggests that total consumption isn't highly dependent on concentration.

We agree that flies consumed more in Ja et al. 2007. There were multiple differences between our methods: they used only males, they included yeast in their liquid food, and their flies were habituated to the assay for 24 hours before the measurements started. Any one of these could have contributed to the differences in total amount consumed.

b) It seems strange that cumulative consumption over 6 hours is a straight line (Figure 1B). Given that these flies were starved, I would expect to see high consumption in the first hour that then decreases substantially. Also, feeding bouts are typically clustered in the morning and evening, and not distributed equally throughout the day. Any thoughts on this?

We agree with the reviewer that normal feeding behavior occurs at dawn and dusk. However, these flies were starved and then placed into a new environment (FLIC assay). We believe that given 24 hours, they would have habituated to the environment and likely would have settled back into their typical circadian rhythm of feeding.

c) It seems strange that changing the sugar concentration more than 10-fold results in hardly any difference in feeding (Figure 1D-F). I might expect no difference if measuring over an entire day when homeostatic processes dominate over taste, but this is a 20 min assay. Also, the authors use the lack of difference in bout number between mated and virgin females as a justification for why feeding initiation (PER) is unlikely to be different. However, the bout number is also similar across sugar concentrations whereas PER is very different (at least for fed flies), which casts doubt on that logic.

It is possible that 20 minutes is not enough time capture the differences between concentrations given the amount of noise in the data. With longer assay duration and higher n, it is possible that we would see a more homeostatic process where flies consume less when exposed to the higher concentrations of sugar compared to those exposed to the lower concentrations.

What is clear and consistent from the data is that mated females consume more than virgins and that they do so via longer drinking bouts.

2. The text states: "Thus, like hungry flies (Itskov et al., 2014), mated females consume more by engaging in longer feeding times rather than by initiating more feeding events." This seems misleading since the paper by Itskov et al. shows a huge difference in the number of sips for fed and starved flies, whereas the sip duration did not differ (Figure 5C and 6E).

This sentence was modified: “Thus, mated females consume more by engaging in longer feeding times rather than by initiating more feeding events.” (LINE 96-98).

3. What was the rationale for focusing on pCd-2a and pCd-2b rather than pCd-2c?

We prioritized making split-GAL4 lines for pCd2-a and pCd-2b since they have a greater number of synaptic connections with upstream SAG and pC1 and well as downstream (see Figure 5A and Figure 6B).

4. In Figure 5 – Figure supplement 1, the driver lines seem to label a few cells other than pCd-2 (e.g. in the optic lobe, abdominal ganglion, or along the surface of the brain), so it seems strange that in most cases the male brains completely lack any expression, even of the non-pCd-2 neurons.

The labeling on the optic lobe and the surface of the brain is not to neurons but nonspecific binding to retinal tissue. The labeling of the abdominal ganglia in pCd-2a females is female specific. The off-target neuron labeled in pCd-2b females in also labeled in males.

5. As noted in Point # 2 of the Public Review, I think that the data show a possible convergence of mating and hunger signals in the same cells, but it hasn't been shown definitively. Given this, I think the title is a bit speculative. I also think this sentence in the abstract is not really supported by the data: "Together, this circuit transforms the mated signal into a long-term hunger signal."

To tone down the language claiming that Lgr3+ neurons integrate these two signals, the following sentence was removed from the discussion: “The sexually dimorphic Lgr3+ neurons are the nexus integrating mating status and neuroendocrine hunger signaling.” The following was added:

“Importantly, it has not been established how Dilp8 influences the activity of Lgr3+ MBNs. Regardless, together these data suggest a proposed model for how Lgr3+ mBN neurons integrate signals from the mating status and hunger/satiety systems (Figure 7). (LINE 278-281)

6. A functional connection between SAG and pCd-2 neurons was not tested here. This is not critical, but it would be interesting to not only confirm this connection but to test how the co-activation of pC1 and SAG affects the pCd-2 neurons. This might hint at why the circuit evolved to have both pC1 and SAG connected to pCd-2 rather than just one or the other. If the authors have ideas on this they could speculate in the Discussion.

The functional connectivity experiments proposed are challenging and beyond the scope of this study. It is interesting that SAG and pC1 synapse onto pCd-2b/c. This may serve to amplify the signal from the mating status circuit to pCd-2 or to allow activation of pCd-2 under circumstances where pC1 is not inhibited. Further research is required to explore this circuit architecture.